# Gesture, Music and Computer: The Centro di Sonologia Computazionale at Padova University, a 50-Year History

**DOI:** 10.3390/s22093465

**Published:** 2022-05-02

**Authors:** Sergio Canazza, Giovanni De Poli, Alvise Vidolin

**Affiliations:** Centro di Sonologia Computazionale, Department of Information Engineering, Padova University, 35131 Padova, Italy; giovanni.depoli@unipd.it (G.D.P.); alvise.vidolin@dei.unipd.it (A.V.)

**Keywords:** music, sound and music computing, human–computer interaction, musical cultural heritage, multimodal interaction, history of Padova University, environment for learning and inclusion

## Abstract

With the advent of digital technologies, the computer has become a generalized tool for music production. Music can be seen as a creative form of human–human communication via a computer, and therefore, research on human–computer and computer–human interfaces is very important. This paper, for the *Sensors* Special Issue on *800 Years of Research at Padova University,* presents a review of the research in the field of music technologies at Padova University by the Centro di Sonologia Computazionale (CSC), focusing on scientific, technological and musical aspects of interaction between musician and computer and between computer and audience. We discuss input devices for detecting information from gestures or audio signals and rendering systems for audience and user engagement. Moreover, we discuss a multilevel conceptual framework, which allows multimodal expressive content processing and coordination, which is important in art and music. Several paradigmatic musical works that stated new lines of both musical and scientific research are then presented in detail. The preservation of this heritage presents problems very different from those posed by traditional artworks. CSC is actively engaged in proposing new paradigms for the preservation of digital art.

## 1. Introduction

One of the first methods of producing music was the use of the human voice. In particular, singing enables the musician to create a direct relationship between her/his ideas and their realization. Soon enough, existing objects were used and, after that, new instruments were invented to produce sounds and music, becoming increasingly sophisticated and expressive. Technology plays an important role in identifying the most suitable mechanisms and materials to generate sounds according to the specific needs of the culture of the time and to stimulate new expressive forms of music. On the other hand, technological innovations for music have often had a significant impact on other fields as well. Musical instruments and their acoustical properties are the medium between a performer’s actions and the sound, i.e., between a musician and the listener. The musical instrument is therefore a paradigm of communication between human beings, mediated by technology. With the advent of digital technologies, the computer becomes a generalized tool for producing music: if music is the message, the computer is the instrument to convey the message. Today, most of the human–human interaction is actually a human-to(-computer-to)-human interaction. Music can be seen as a creative form of human–human communication via computer, and therefore, research on human–computer and computer–human interfaces is important. Thus, research on sensors for acquiring signals from the physical world to be input to computers and actuators for rendering devices of media signals to the public is essential for expressive artistic communication.

Cellphones, ringtones, alarms, automated messages and beepers are included in the modern soundscape. These sounds have little relation to electronic and/or computer music: the technologies are the same, but the content (and the aim) is very different. Moreover, most pop music is produced today using electronic technology. An example is the modern piano man, a technological version of the old “one man band”, which can simulate an entire orchestra with laptop and electronic devices placed on the piano stand or inside a keyboard. Auto-tune (a software for pitch correction and vocal effects) is an example of the “live” (not trivial) use of computer technologies. Even western classical music is often produced today using digital technologies: small audio excerpts are extracted from various rehearsals and painstakingly edited together. The balance of the parts is established a posteriori by reshaping the dynamics according to both the conductor’s and the producer’s wishes. In this sense, the recording of a classical piece is more similar to the making of a film than the recording of a theatrical performance. These examples relate to the use of technology (analog or digital, offline or live, at the concert) to imitate and substitute traditional processes and instruments. Electro-acoustic research music, on the other hand, was conceived to realize new musical forms and expressions by means of new technology—forms and expressions that were not conceivable or imaginable with traditional musical tools [1]. The authors use the term “*electroacoustic music*” to include electronic music, computer music, tape music, and live electronics.

This work for the Special Issue of *Sensors* dedicated to the 800th anniversary of the University of Padova intends to be a review of the research in the field of music technologies at Padova University, focusing on the interaction between musicians and computers and between computers and the audience.

The first attempts to use electronic technology to produce music at the University of Padua date back to the late 1950a, when an innovative photoelectric organ was designed [2]. In the late 1960s, the group *Nuove Proposte Sonore* (NPS), led by Teresa Rampazzi, introduced analog electronic music (Figure 1) in the Conservatory of Music in Padova and then began collaborating with the university in the field of computer music. In the early 1970s, a group of researchers and musicians started working on speech synthesis and computer music [3,4]. In 1979, these activities were institutionalized by the university with the creation of the Centro di Sonologia Computazionale (CSC) (Figure 2). Since its founding, CSC has been a leading research center in the field of computer music. Musical creativity has been a stimulus to pursue new paths in scientific and technological research. Over the years, CSC’s research has addressed several topics (described in detail in [5]), which include sound processing, expressiveness, multimodal interaction, and musical cultural heritage. On the other hand, theoretical and scientific achievements have been constantly applied to music production. More than two hundred musical pieces have been made at CSC and have received widespread recognition (http://csc.dei.unipd.it/multimedia-works/, accessed on 15 March 2022).

The article is organized as follow. In Section 2, the main concepts regarding music and gestures are presented. Next, Section 3 accounts for the research on interfaces and interaction developed at CSC, along with some relevant applications. Section 4 considers some achievements in music production and describes several important musical works along with the reasons behind them and the technological innovations achieved. Finally, Section 5 casts a look at CSC’s visions of the future, built on the basis of its history.

## 2. From Gesture to Music to Audience

### 2.1. Music and Gesture

Music is the result of human gestures performed on special instruments to generate and control the production of musical sounds. Except for the use of voice, in which sounds are produced internally by the human body, when a musician strikes a resonator—plucks a string or blows into a tube—s/he excites a sound generator. Conversely, the variation in the length of the string or of the air column that oscillates inside a tube are examples of gestural control of a particular musical sound. Therefore, when using a traditional musical instrument, the performer’s gesture provides the energy for the generation of the sound, the control of the sound type, and its evolution in time. Often, the generation also includes control gestures, and generally, music can be considered as produced by the coordination of parts of the human body: hands, mouth, feet, etc. [6].

With the advent of electrical and, later, electronic technology, sound generation is derived from electrical energy and the control of the produced sounds is carried out by electrical signals. It is then possible to separate the sound generating function from its control, which in the past had only happened in pipe organs. Already in the third century BC, an instrument was built in which the compressed air to be sent to the organ pipes was supplied by hydraulic energy. The human control gesture, therefore, must be translated into one or more electrical control signals using sensors that allow detecting the human gesture in its various dimensions.

In this context, a variety of transducers were developed to convert the complexity of the human gesture into an appropriate number of control signals. The first transducers were one-dimensional, i.e., able to translate the gesture in a variation of a single parameter of the electrical signal (on/off, amplitude, frequency, etc.), but soon, researchers stated to develop sensors for more sophisticated control of music. In the early years of the last century for example, capacitive sensors (devices that can measure changes in electrical capacity, used as proximity sensors) succeeded in the gestural control of pitch and dynamics in the theremin that can be controlled by the movement of two hands in the air. These fascinating innovations, however, soon clashed with musical practice since it requires many years of study on the instrument to make it become a sort of musical extension of the player. Therefore, most manufacturers of electrophonic instruments chose to use the organ keyboard as the basis for controlling sounds and musical performance, leading to several generations of electronic keyboards increasingly rich in new sounds and effects [7]. This approach, however, has proven inadequate for electroacoustic music, so an area of research and creative practice has been established to develop new interfaces for musical expression and to seek new paradigms for instrumental performance [8].

### 2.2. Gestures and Music Instruments

The CSC has been researching digital sound generation since the 1970s. In its early days, music was produced offline by mainframes. In batch computer music, the only way to input data to the machine was through alphanumeric symbolic languages that represented the synthesis/processing algorithm and the sound parameters. Each expressive intention needed to be explicitly formalized in the input data. The performer’s gesture was not taken into account: the composition of the sound and the composition with sounds replaced the traditional performance with instruments. In the 1980s, real-time music systems started to be designed, which allowed an increasingly effective interaction between the performer, the machine and the listener and fostered new performance practices. Interaction and music performance became one of the focal points of CSC research.

In this section, we analyze the architecture of a music performance system, discussing the issues of input and output devices, and the intermediate layer for mapping information to the processing unit.

#### 2.2.1. System Structure for Music Performance and Interactive Multimedia

In many acoustic instruments, the playing interface is part of the sound generation mechanism. Since they are intrinsically linked, their relations are complex and determined by physical laws. In electronic and digital instruments, the physical interface—or control device—is disjointed from the sound generation and processing system. This offers the designer and the musician the possibility of establishing the correspondence between the interface and the generation system that best suits their musical ideas, as well as to vary it according to their own expressive needs. In this way, the complete behavior of the instrument can be changed.

It is worth distinguishing the interface—or controller—from the sound generating system by including a *mapping* layer interconnecting sensed parameters from the controller with input parameters of the synthesis engine. While the sensors and input devices determine the mechanical and acoustic constraints of the system and the sound engine affects the sound characteristics, the mapping defines the performative qualities of the instrument [9].

We will use the term *Music Performance System* (MPS) to indicate a set of hardware and software components that mediates the musical message between the performer and the listener. Such a system (Figure 3) typically consists of the following components [10]:Gestural controller—the input device for musical control in which physical interaction with the player takes place and is made by one or several sensors and a data acquisition subsystem to capture performer gestures;Mapping layer—software subsystem to translate gestural data into musical and sonic information;Compositional layer—a possible structural layer for controlling the organization of sounds and compositional processes;Sound generation unit—real-time audio generation and processing engine in which sound synthesis parameters are modulated by live input;Output system—audio output subsystem that realizes the projection of sound in acoustic or virtual space.

The approach pioneered by music was extended in the 1990s to other art forms creating *interactive multimedia*, a term that refers to the simultaneous and synergistic use of different types of communication channels, such as texts, sounds, images and videos. This was made possible by digital technologies that allow information from multiple sensory channels to be treated uniformly. As a matter of fact, different modalities such as these, which are traditionally on different supports and with different objectives, have in common the same foundation—being information represented in bits—in the digital domain; therefore, they share the same information processing and transmission technologies. We are witnessing, on one hand, the unification of conceptual elements and languages previously expressed through different media in the digital medium and, on the other, the strong integration of different expressive codes enabled through digital coding and IT tools. In the modern concept, multimedia are no longer conceived of as a juxtaposition, the placing together both synchronically and diachronically of different media related to sound, acting, decor and lighting, but rather as a micro-integration, or close linkage, of different media. This has greatly favored the creation of new artistic typologies that integrate various forms of communication into a single object, or artwork, and that experiment with several media technologies that are gradually developed. Each technology facilitates the investigation of new expressive possibilities. The sector that encompasses such experiences is called *media art*.

Due to the availability of digital technology (sensors, computers) and software applications, a whole new area for artistic exploration has become easily accessible and the interaction between multimodal experiences and multimedia technology plays a key role. In the performing arts, multiple sensory modalities, including auditory, visual, haptic and tactile, as well movement perception, are often involved in the communication between the artist and the audience [11]. Thus, the concept of an MPS can be generalized as a *Multimodal Performance System* or multisensory system, which mediates the artistic message between the performer and the listener and is based on an intelligent human–machine interaction (Figure 4). It is worth noting that the performer and the user may be the same person in interactive multimodal systems. Thus, the last two components of an MPS, as seen above, can be generalized as:Signal processing unit—real-time audio/visual generation and processing engine in which synthesis parameters are modulated by live input;Actuator system—a rendering system in the modality of choice, i.e., an interface to communicate the digital message to the audience.

Finally, a *performance environment* is the musical interface (hardware and software) that allows for transforming a complex technological system into a general musical instrument that can be used by the interpreter when playing a specific musical composition. The planning of any such environment is left to the interpreter as to reconcile the personal performing style with the characteristics of a single work [12].

#### 2.2.2. Input to the System

The input device usually consists of a (contact or non-contact) sensor system that captures the performer’s physical gestures and produces digital data representing the musician’s actions (e.g., 3D camera, IR camera or, if more speed precision and sensors needed, systems such as PhaseSpace motion Capture) to control the MPS. Sometimes, physiological signals (including Electroencephalography systems, Functional Magnetic Resonance Imaging, Near InfRared Spectroscopy systems, eye-tracking systems, response devices EMG or muscle tension) are used as a source of information. In particular, recently, brain–computer interfaces have received increasing attention in music applications.

The audio itself can be a source of information for controlling the system, such as detecting acoustic events or locating the sound source. In the case of music, the natural sound of an acoustic instrument can be a rich source of data to be sensed. Information on the actions and gestures of the performer can be derived by the acoustic analysis of the structural properties of the sound produced by the instrument. The input system acts as a machine listening device. In this way, the music instrument becomes an input device and can be used by the performer without losing its traditional performance practice.

The *signal processing unit* of the MPS should not be only considered as a signal generation device but also as a sound transformation device. Audio can be direct input to the system in the case of live electronics, a field in which CSC has conducted a lot of research and musical experimentation. *Live electronics* is a musical practice where sounds are processed live with an electroacoustic system; acoustic and electroacoustic sounds are present simultaneously; the electroacoustic system becomes an extension of the voice and/or the instrument, i.e., the interaction between the acoustic and electronic performer. As the conductor during a concert controls the execution performed by different musicians using gestures, so the interpreter of live electronics processes the audio signal produced by other musicians through specially programmed computers and suitable human–machine interfaces during the performance.

#### 2.2.3. Information Mapping for the Processing Unit

The *mapping* component refers to the link strategies between the outputs of the gestural controller and the input parameters of the sound generation or signal processing algorithm. The simplest strategy is *direct mapping*, i.e., using one gesture parameter as the input parameter of the algorithm. An example is one-to-one mapping, in which the position of the finger on the keyboard directly maps to the frequency of the note played, and the velocity of the finger hitting the key maps to the loudness, or in which the value of a potentiometer is used to modify the center frequency of a filter, or, again, Key velocity is used as input impact velocity of the hammer on the strings in a physical model of a piano. However, in instruments such as violins, the bowing gesture offers a high degree of continuous control and affects many sound generation parameters, and more complex mapping strategies should be devised. It is advisable to decouple input gestural data from processor control data in two (or more) steps through *layered mapping*, which includes an intermediate representation of parameters. The intermediate information can be at different representation levels, such as physical, perceptual and even conceptual. Careful theoretical, scientific and musical research on this intermediate level has been at the base of many of CSC’s scientific and musical achievements (see e.g., Section 3.2.1).

#### 2.2.4. Output from the System

The *actuator* component refers to the system used to convey the computed signal to the audience. Loudspeakers and headphones are the main devices for audio signals. Some composers also use the acoustic musical instrument, augmented with suitable transducers, to diffuse the computed sound, such as in the piece *Altra voce* by Stefano Gervasoni (see Section 4.6). Concert spaces, with their resonant and acoustic properties controlled by sound localization/spatialization algorithms, can also be conceptually considered part of the actuator system (see e.g., *Prometeo, tragedia dell’ascolto* by Luigi Nono, in Section 4.2). Screens, video projectors and spotlights are used to render the visual element and haptic devices to convey touch and vibratory sensations.

To perform well, the musician needs *feedback* from the gestures s/he is making. Feedback can take several forms, but in essence, it is information that gives the musician an indication or confirmation of what s/he has articulated just then. Feedback is the visual, auditory or tactile-kinesthetic mechanism by which a performer senses the state of her/his or her instrument [13]. The successful integration of more than one of these modes (e.g., tactile, sonic) contributes to the creation of a clear understandable coherence. An acoustic instrument inherently provides performance feedback to the player in the form of vibrations and a perceivable instrument state. An electronic instrument often has no feedback mode except for the sound produced. While feedback is first apparent to the musician, giving him a point of reference for what s/he is doing, the action/reaction feedback dynamic should also be apparent to the audience. This gives the listeners the key to following the expressive space that the performer is creating [14].

## 3. Scientific Research for and Inspired by Music

CSC research on interaction focused on IO interfaces for creating and preserving musical cultural heritage, communicating and mapping expressive content, and multimodal interaction for learning and well-being.

### 3.1. Interfaces for Creation and Preservation of Musical Cultural Heritage

In early computer music, real-time interaction was not possible. In the 1980s, live computer music performance began to take place on special-purpose synthesis hardware developed by researchers and industries. Later on, software synthesis became possible on personal computers. Since the act of playing consists essentially of producing a sound and continuously controlling its characteristics in timing and in timbre, a performance depends mainly on the degree of interaction between the player and the instrument. Musicians felt the need for new interfaces for music expression, which would allow a strong exchange of information between the player and the system for control and feedback and between such system and the listener [15]. Here, we will present CSC research on interfaces for the creation and preservation of musical cultural heritage, expressive information representation and processing, and multimodal systems for learning and well-being.

#### 3.1.1. Interfaces for Creation

Interactive musical instrument development in the 1980s focused on novel controllers. Originally, such interfaces, which converted analog control voltage output of sensors to digital data, were the fruit of specialized design efforts at early computer music centers, such as CSC. Later advances in technology led to the emergence of new music controllers from commercial instrument companies. Finally, general-purpose sensors and controllers started being used in music production, and the research effort was then devoted to information fusion and mapping.

In this section, speaking of inputs, we present some types of input devices developed by CSC in the early period and how to use audio to extract gestural information, along with creative applications that take advantage of such an approach. Regarding outputs, we present CSC research on haptic feedback and how to render 3D sound in space. CSC created innovative interfaces for playing instruments, which aimed to control timbre and virtual space, polarizing the interest of composers who wanted to simultaneously control multiple parameters, synthesis algorithms and sound spatialization. In Section 4, we will illustrate several significant musical works in which controllers are used in innovative ways.

##### Augmented Instruments as Controllers

An acoustic instrument can be augmented by sensor technology and signal analysis techniques enabling user interactivity and control capabilities of digitally generated or processed sounds. In this way, the performer expands the gestural control palette while preserving traditional playing techniques and exploiting the pre-existing sensorimotor skills. The concept of valve augmentation was central to the design of the *Meta-Trumpet* controller by Impett [16]. The trumpet valves, an existing mechanical interface of the acoustic instrument, were equipped with sensors for detecting the valve’s movements. Information about the player’s physical contact with the trumpet was provided by two pressure sensors, mounted on the right of the third valve casing, and two mercury switches, together providing a four-stage value for the left-to-right inclination of the instrument. The trumpet valves contained a shielded magnet inserted into the cavity at the bottom of the piston. The resulting magnetic fields were read by Hall effect sensors fitted in specially built extended lower valve caps. The valve positions can be used by the player to control sound synthesis or be interpreted as musical parameters abstracted from the act of performance.

Moreover, the *Meta-Trumpet* uses ultrasonic distance sensors, accelerometers and tilt switches to measure position, motion and rotation, respectively, at the tip of the trumpet bell. In this way, the ancillary gestures of the performer are acquired and can be used as gestural control.

##### Alternative Controllers

Normally, in order to play an instrument, the musician must have free hands, and as we generally have greater control of our mouth shapes than other body parts, the mouth could represent the more obvious choice. Moreover, the musician does not need to be distracted with additional activities such as speaking, making sounds or simply blowing (as with the breath controller); s/he only has to change the inside geometry of her/his vocal tract. Orio [17] designed a new gesture interface that reveals and maps postures of the oral cavity; the user only needs to alter the inside shape of their mouth for the changes to be sensed (Figure 5). The interface emits white noise inside the oral cavity and picks up the filtered signal with a two-acoustical waveguide system. The oral cavity filter parameters are estimated with LPC analysis of the recorded signal. Then, the Principal Component Analysis extracts up to three independent mid-level parameters used for control. The interface needs a short training phase in which a new user adopts all the mouth postures s/he wants to use for the control. Even if the interface was designed for musical purposes, it can have applications in other fields, such as a mouse for people with disabilities or an aid tool in phonetics.

In interactive multimedia installations (IMIs), the viewer’s body becomes the interaction device. CSC developed many IMIs that required special input devices to detect the actions and positions of a visitor. For example, the goal of the interactive sonification of a painting of Hartwig Thaler was to provide the user with an immersive experience of a soundwalk simulating an imaginary navigation inside the painting (Figure 6). The controller was designed to be an integral part of the artistic installation. A wooden runway, equipped with piezoelectric sensors to detect the user’s position, activates the corresponding soundscape according to the visitor position, which results in a sort of three-dimensional spatial development from the original two-dimensional artifact [18].

##### Audio as Sensor

The experience gained by the CSC in the analysis and synthesis of audio signals led to the development of several methods of extracting information from the acoustic signal. Two approaches used for artistic creation are presented below. Innovative audio-based systems for surveillance and safety in the workplace will be presented in Section 3.4.

**Detecting Detecting guitarist’s gestures** Typically, in HCI, gestures are detected with position, pressure, or motion sensors. Orio [19] proposed a new method for sensing the musical gestures of a guitarist: the actions of the performer, with their nuances, are recognized by signal analysis of the generated sound. With a classic guitar, the position in which the guitarist’s left-hand fingers press the string can be determined by pitch analysis, and the strength by which the string is plucked by right-hand fingers can be revealed by the loudness. The way a string is plucked and the small gestural nuances used by the performer to convey its expressive information about feeling and intention to the listener produce differences in timbre, which the proposed model is able to detect. A group of guitar sounds played by changing the plucking techniques in several ways were analyzed using different methods, such as time-frequency spectral analysis, perceptual parameters such as as brightness and spectral irregularity and mel-frequency cepstral coefficients (MFCCs). The possible variations in guitar timbre were highlighted in relation to the different plucking gestures. The model was able to classify the plucking technique by comparing the incoming sound with the analyzed repertoire.

***Voice Painter*****application** Research activities conducted within the European Network of Excellence “Enactive Interfaces” aimed at developing new ways of mediation between humans and the world through digital technologies.

Enactive interfaces are interactive systems that allow the organization and transmission of knowledge obtained through perception–action interaction in the surrounding environment. To explore whether the auditory and musical experience was prone to such kind of interaction, CSC developed an enactive system (*Voice painter*) that stimulates the user-perceiver-enactor to use her/his body and in particular her/his voice to create a painting and discover the relation between her/his actions and the signs that s/he can produce. The central idea was to exploit the most relevant features of vocal expression and map them into graphic features, thus creating a simple and versatile instrument that can be used by an experienced performer as well as by a naive user. The mouth acts as a brush: in order to draw on the entire screen surface, the user is forced to move, therefore involving the whole body rather than just the voice as an input instrument (Figure 7). The user position, tracked by a microphone array, controls the rendering place on a canvas. Some prominent voice features are mapped into well-recognizable graphic features with different color gradients and sizes. The resulting visual effect is a sort of abstract sketch that can contain well-defined geometric elements produced by short voiced segments and/or particle-like signs due to unvoiced segments. In this way, the user has a natural interaction with the virtual canvas since the visual feedback is temporally and spatially correlated with the voice event. The action–perception closed loop is then recreated.

Painting is an example of autographic art, where there is a complete product in se at the end of the creation process; music instead is an example of an allographic art (i.e., an art performed with the contribution of multiple actors, such as composer and performers). The distance between these two forms of art can be dramatically reduced by interactive cross-modal performance systems [20], where, for instance, a painting can be seen as the result of a live performance. The *Voice painter* was born with artistic applications in mind: to create an instrument that could mediate between allographic and autographic arts, allowing sophisticated performances based on a musical notation (used then to paint) as well as improvisations or simple entertainment. It offers a new artistic metaphor as well as a potentially useful tool for speech therapy programs [21].

##### Feedback to the Player

In traditional instrument performance, the relationship between the musician and her/his instrument is highly developed and deeper than what we expect in typical human–machine interaction. Many years of practice are needed to build up an intuitive feeling of the instrument. Creating such an intimate dynamic performer/instrument relationship is a central issue in new computer instrument design. Of the factors that contribute to helping the musician achieve this, feedback is an essential component. When playing a musical instrument, a player receives as feedback not only the generated sound but also the haptic interaction, arising during the contact between player and instrument. Such haptic interaction stimulates several senses in the player: tactile, kinesthetic, proprioceptive, etc. It constitutes a bidirectional communication channel between player and instrument and helps to attain an intuitive musical fluency.

These considerations led to various research activities aimed at creating haptic displays for music [22], pioneered in the landmark paper by Cadoz et al. [23]. The CSC has contributed to this direction with two main projects: one by designing a haptic musical interface that allows a force feedback reproducing the real mechanism and the second by studying the performer’s perception of the vibrotactile feedback added to virtual pianos.

**Force feedback** From the physical point of view, the player exerts a force (by hand or mouth) on some parts of the instrument, exciting the sound generation mechanism, and the instrument reacts toward the player with a force. This feedback contains useful information on the instrument behavior and can be exploited by the player to adjust the gestures to obtain the desired tone nuances. This strong correspondence between generated audio and touch feedback, however, is lost in many digital keyboard instruments, in which sound generation is controlled only by key-velocity (how fast the key was pressed) and after-touch (the amount of pressure on a key once already held down) MIDI parameters. In these keyboards, the touch feedback does not depend on the instrument being simulated. Real keyboard instruments (such as harpsichord, piano, or pipe organs) are characterized by very different excitation mechanisms and thus give quite different touch (force) feedback to the player. The lack of this precise haptic feedback is a severe obstacle to controlling the gestures the musician is accustomed to and limits her/his expressive possibilities.

To overcome this problem, a multi-instrument active keyboard (MIKEY) was designed (Figure 8), in which the actuators connected to the keys allowed the reproduction of the touch feedback that a user would experience when interacting with the real instrument being emulated by the synthesizer. The force generated by a direct-drive low-friction motor is computed in real-time by a simplified but effective dynamic model, which simulates the mechanical components and constraints of the grand piano, harpsichord and Hammond organ action mechanisms [24].

The system allows the pianist to detect the escapement mechanism’s re-triggering by paying attention to the interaction force between keys and fingers, originating during the descent of the key.In this way, s/he could adjust the key motion to obtain the fastest repetition of the note.

**Vibrotactile feedback** In the piano, there is not only the direct feedback produced by the kinematics of key movement and escapement mechanisms re-triggering but also the indirect one produced by string vibrations and perceived through the fingers as a vibrotactile feedback that follows the initial keypress and the related early feedback during piano playing. Fontana et al. [25] investigated the hypotheses that string vibration at the keyboard is perceived through the fingers and that the same vibration affects the perceived quality of the piano. The vibrotactile feedback was conveyed by two tactile transducers attached to the bottom of a digital piano keyboard (Figure 9).

The experiment showed that sensitivity to key vibrations is highest in the lowest range and decreases toward higher pitches, and the addition of vibrotactile feedback to a digital piano significantly modifies the performer’s preferences while playing and the perceived quality of the instrument.

##### Spatialization and Binaural 3D Audio Rendering

Several composers deem the location and movement of sound sources to be a compositional parameter that is fundamental to defining the musical structure. Spatial audio technologies allow disjoining the perceived location and distance of the sound source from the actual one (loudspeaker of headphones). Prompted by composers’ wishes, CSC developed innovative techniques for spatial audio rendering with a focus on binaural (i.e., headset-based) audio processing and real-time 3D audio rendering. With headphones, the listener’s perception cannot exploit the pinna and torso reflection properties, which are highly individual. In this way, the vertical coordinate is lost and should be recreated by filtering the signal by the so-called HRTF filters, which depend on the direction and are highly individual. A structural model for personalized HRTF was developed [26] allowing a real-time simulation of multi-source acoustic environments in which the spatial positions of both the listener and the sound source can dynamically vary [27]. The rendering of the filtered signals through motion-tracking headphones (Figure 10) allows the user to perceive the position of the virtual sound sources independent of their movement [28].

The main results of this project concern binaural rendering through models and algorithms with low computational requirements, which integrate techniques for personalization and adaptation to users’ anthropometry.

#### 3.1.2. Audio Documents Preservation

Digital archives and libraries are increasing their online presence. This phenomenon represents a crucial impulse for the development of cultural and augmented learning tools. A widespread and easy dissemination of today’s musical culture should be granted as a democratic act for the sake of future generations. Thus, international archives and library communities are faced with a key challenge: the creation of new systems for the acquisition, preservation and transmission of information [29,30].

The preservation of electroacoustic music heritage presents very different problems from those posed by traditional music. To safeguard such heritage, not only the acoustic information recorded on tape must be digitized but also the performance praxis must be preserved. In this sense, it is necessary to preserve all the elements that allow us to understand and reconstruct the set of work production processes, from the specific details of the composition to the technological system and operating practices used [29,30,31]. The performative or behavioral aspects are rarely documented in detail, and when they are, the documentation only refers to the technology of the time in which the work was produced. Often, only the author’s notes and annotations in her/his personal archive are available. Therefore, every effort must be made to keep and make accessible all information relating to the work, the equipment used and the performance practice. To this purpose, it is very important to properly address the issues of personal archives preservation and access [32,33].

**Artificial Intelligence for musical cultural heritage** Several carrier (magnetic tape, film, paper) alterations, such as blocking, leafing, windowing, spoking or embossing, fungi and molds may be detected in an initial visual inspection of the original artifact (analogue). At this first stage, an expert technician could assess whether the carrier needs to be restored before being subjected to the second stage, the digitization process. However, some kinds of corruption or discontinuities could only be detected during digitization. Consequently, both stages (visual assessment and digitization) are essential for the evaluation of the document preservation condition. The carrier alterations (or discontinuity or irregularities) from its original manufactured state are visually detectable. Irregularities included in the methodology defined at CSC are: splices, brands on tape, ends of tape, damaged tape (ripples, cupping, damage to tape edges, dirt, biological contamination), marks (signs or words written on the back of the tapes, see Figure 11) or shadows.

The *Video Analyser* software developed at CSC aims to detect the significant frames from a digital video. The detection is focused on two areas: the first one is on the reading head of the tape recorder, while the second one is under the pinch roller. The *Ends of tape* and *Shadows* irregularities are detected by evaluating the pinch roller position change. The others are evaluated on the reading head area. Significant frames are extracted comparing consecutive frames. If there are important color changes between the two frames, the last one is indicated as significant. The implementation is based on OpenCV libraries. The CSC proposes an algorithm that classifies the images provided by the video analyzer using the neural network GoogLeNet, Keras libraries and TensorFlow.

Audio corpora, which was used to train the AI, came from several preservation projects in which CSC had participated. In Pretto et al. [34], it is possible to find the list of such projects and a more exhaustive description of the related results and statistical data. The obtained AI accuracy ranges from 70% to 100%, where differences depend on the different carriers used and on the presence of discontinuities.

Whilst being the first attempt at applying an AI approach to audio preservation, the high accuracy score proves its potential. These tools aim to help the operators involved in the digitization process by providing specific software able to assist in technical decisions related to the playback device configuration and to validate the results of the process. From the musicological point of view, automatic tools that detect discontinuities in audio documents will be a powerful aid for a correct and complete analysis of musical works.

**Photos of Grooves and HOles, Supporting Tracks Separation (Photos of GHOSTS, PoG)** All the most important libraries in the world widely use automatic text scanning and OCR technologies for text recognition to preserve the original document. Such care does not apply to audio preservation, and often, the Analog/Digital (A/D) transformation of historical sounds remains an invasive process. PoG, a new innovative system developed by CSC, could change that. PoG can reconstruct the audio signal from an HD image of a shellac disc surface (Figure 12).

A shellac disc is a widespread audio mechanical carrier in which the recording is captured on the disk surface in grooves carved by a sound-modulated stylus, both directly (in case of acoustic recordings) and emitted by electronic amplifiers. Millions of shellac discs are in audio archives around the world (R&B, Jazz, Ethnic, Western, classical, to name a few genres), and they were never re-recorded.

In the literature, there are a few approaches to using image processing to reconstruct sound. PoG is characterized by the following features:It is able to recognize different rpm and to perform track separation automatically;It does not require human intervention;It works with low-cost hardware;It is robust with respect to dust and scratches;Its output is made of de-noised and de-wowed audio using restoration algorithms.

PoG stores hundred of equalization curves along with their own references and parameters (date, company, roll-off, turnover) that could be applied during the audio reconstruction process. Moreover, using consolidated tools for iris detection, PoG detects the disc center and radius and automatically rectifies grooves and separates tracks. From the light intensity curve of the scanned image, PoG models the grooves and collects the audio samples [35]. PoG is the base strategy for:Large-scale (massive and semi-automatic) A/D transfer of mechanical recordings, retaining maximal information (2D or 3D model of the grooves) about the native carrier;Small-scale A/D transfer processes, where there are not sufficient resources (trained personnel and/or high-end equipment) for a traditional transfer using turntables and converters;Active preservation of carriers afflicted with heavy degradation (breakage, flaking, exudation).

#### 3.1.3. Interfaces for Simulating the Original Listening Experience of Audio Documents

The access methodology to audio documents should aim at a faithful reproduction of the audio content and also at a simulation of the original listening experience. Since electroacoustic music on magnetic tape is strongly linked to the physical medium and the recording system, the virtual simulation of the peculiarities of the device and its use is a necessary step to preserve the original listening experience. Traditional access methods, such as the CD-A edition or listening through software jukeboxes, are inadequate to restore the listening experience of old analogue media, as they do not respect the characteristics of the original document, nor do they provide adequate access methods to metadata and ancillary information.

A skeuomorphic user interface (UI) is a way in which users are presented with information on an electronic device that mimics the appearance and behavior of a physical artifact. As they accurately replicate the feeling of the interaction with an object, skeuomorphic UIs have been useful in recreating past musical instruments and artifacts no longer available in tangible forms. This is the case of tape music fruition: the original tape recorder can be moved into its virtual form and installed on a mobile device. A mobile device can be carried around easily, and the presence of a touch-based UI provides the user with a richer, more accessible and more faithful experience, as the virtualized tape recorder can be easily manipulated. Moreover, the multimedia features of modern mobile devices support the display of all the non-musical information related to the musical piece to further enhance fidelity.

The reproduction of the appearance, commands (real-time modification of the speed and equalization) and moving parts of the tape recorder (running of the tape) are important in order to make the virtualization of the playback tool complete. CSC [36] has created several software applications for mobile devices and for web access for the most popular analogue sound documents readers of the 20th century (namely, tape recorders, gramophones, turntables and old equalizers).

In the REMIND mobile app shown in Figure 13, as in a real tape recorder, it is possible to interact with different levers to adjust the music reproduction parameters, such as tape speed and the equalization curve definition. The REWIND web app in Figure 14, on the other hand, is the virtualized version of a gramophone and an Albiswerk equalizer (an equalizer largely used in the Studio di Fonologia della RAI di Milano, one of the most important centers of electroacoustic music in the world during the 1950s and the 1960s). The REMIND and REWIND apps could deal with all sorts of metadata information related to a musical piece, both textual ones (e.g., author, year, country of origin) and multimedia ones (e.g., pictures of the original media, video of the original tape running in sync with the audio).

The main contribution of this project is a new, robust and philologically informed (i.e., one that considers the cultural context) methodology for accessing historical audio documents, where realism in the interaction enhances the user experience. The proposed approach can inspire tools for accessing historical documents of other artistic forms, such as cinema, where the working method was also similar to that of tape music.

#### 3.1.4. Multimedia Interactive Installations for Museum Exhibits

Modern museums are increasingly focusing on the visitors rather than the traditional view of focusing on the collections, which has led to a shift in attention from pure preservation to a more public-oriented mission. Their social role has evolved toward active engagement and participation of the public in the process of transmitting knowledge. New tools for the interpretation of displayed objects are required. Interactive and immersive installations are increasingly used in museum and exhibitions to encourage and process audience’s participatory behaviors. They allow new forms of education and experience, where visitors observe and perceive artifacts by acting, and through their multiple senses, a richer and more engaging way of learning than one based on texts and images is obtained.

Musical instruments are both aesthetic and functional objects. Communicating their function and the performer–listener relation is particularly difficult, especially in a museum situation. It is not enough to listen passively to the sounds and the music the instrument produces because it lacks physical engagement. To appreciate its multi-sensory aspects and physical qualities, playing a physical or virtual replica of the instrument itself would allow understanding the object in its totality and to experience its subtle playing techniques in a process that fully engages the visitor. The performance aspect of the instrument–sound relation and the emotional aspects linked to gestures should also be taken into account and communicated to visitors.

To this purpose, in the 2010s, the CSC defined a methodology for designing interactive multimedia installations to present old musical instruments (acoustic and electrophones) in museum settings without losing their cultural context. An example of such an approach is the installation conceived by the CSC for the Music Instrument Museum in Milan, where the original electronic devices of the Studio di Fonologia Musicale (RAI, Milan, Italy) are conserved and exhibited. The Studio represented one of the European centers of reference for the development of electroacoustic music and radically changed the way of producing music and listening to it. Despite the technology involved, electroacoustic music of that time had a non-obvious gestural component. To transmit the experience of how such music was conceived and created, the installation comprises a tangible copy of the devices, which replicates their physical appearance and behavior through a virtual analog simulation of the internal electronic components. In this way, visitors can feel the original experience of playing and producing early electronic music by acting on switches and knobs of the installation [37].

An antique Pan flute from Egypt, approximately dated back to 700 A.D., is exhibited at the Museum of Archaeological Sciences and Art (MSA) at Padova University. An interactive installation that would enhance the viewing experience of the antique Pan flute and understanding the many different aspects of the instrument, such as its history, its myth, its manufacturing and—most importantly—its original sound, was designed.

The interaction is based on two different systems: exploratory and creative. The exploratory interaction carried out on a touch screen allows visitors to move in a multimedia space, which contains all the iconographic and educational information about the instrument. The creative interaction, on the other hand, provides access to the reconstruction of the original sound peculiarities of the instrument using the typical way of interacting with the flute: breath. The visitor can then play the Pan flute by blowing as it was played originally. The experience of a player from the past is somehow preserved and presented to the visitor. An out-of-scale, stylized representation of the 14 pipes of the Pan flute with slits of different lengths were carved on the surface of the cabinet (Figure 15, left). The interaction takes place through sensors (electret microphones) positioned on the holes that represent the mouthpieces of the stylized pipes (Figure 15, right). The user’s blown breath controls the nuances of the digital sound generation. Inside the stylized pipes of the flute, white LED strips light up and vary their brightness according to the intensity of the blow and which pipe is being played. In this way, the visitor can then ”play” the installation by blowing on the pipes, listening to the sounds and looking at the interaction varying the brightness [38].

In the field of museum exhibitions, especially the interactive ones, the design methodology is often specific to the final purpose of the exhibition itself. The main contributions of this research project are the interaction model with the virtual counterpart of the artifact that leverages on multisensory (visual, auditory and tactile) interaction and the proposal of a complete general-purpose design procedure that allows the integration of different disciplines and expertise (engineering, history and archaeology) and different actors (researchers, the museum stakeholders and the visitors).

### 3.2. Expressiveness Communication

Emotions play an important role in human communication and decision making. Understanding and modeling expressive content and emotion communication is important in many engineering applications. Since the 1990s, human expression and expressive behavior have become domains of intense scientific study from an engineering perspective. Research on affective computing deals with computing that relates to, arises from or deliberately influences emotions [39] and has found relevant applications in the field of HCI. Automatically recognizing and responding to a user’s affective states during interactions with a computer can enhance the quality of the interaction, thereby making a computer interface more usable, enjoyable and effective. In Europe, art is often selected as the main application scenario since it is a field where the analysis and synthesis of affect and expressiveness is of central importance. Making machines useful in artistic contexts that rely on different sensory modalities implies that the often subtle nuances of artistic expression should be dealt with in these different modalities. This, however, requires a technology that focuses on affect, emotion expressiveness and cross-modality interactions.

Since expression is an essential aspect of music communication and performance, CSC has been researching the theoretical and practical aspects of expressive interaction. In this context, a CSC research goal has been to create a novel perspective in performing arts by introducing new tools allowing an extension of the artistic languages by acting on the communicated expressive content through technology.

Compared with previous stages in the history of multimedia and music, the main novelty of modern digital technology concerns the encoding, exchange and integration of information using different levels of description. Below, we will present a conceptual model for the representation of expressive information that is effective for mapping in a multimodal context and our research on extending the concept of expressiveness, including sensory aspects, which can be valuable in advanced interaction paradigms.

#### 3.2.1. Multilayer Conceptual Framework

Many human movements in music contexts are called *expressive gestures* and are not directly intended for sound production but rather to convey information relating to the affective/emotional domain. Expressiveness in gestures is conveyed by a set of temporal/spatial characteristics that can be detected by suitable sensors and used to control the musical instrument, possibly combining different modalities [40]. More generally, artistic environments are a good test bed to investigate the expressive aspects of non-verbal human communication and cross-modal interactions.

In the European Union project Multisensory Expressive Gesture Applications (MEGA), the CSC, in collaboration with other centers, has been researching how expressive information can be extracted and communicated in multimodal environments. To this purpose, a *multilayer conceptual framework* for expressive information processing was developed, and cross-domain mapping strategies from gestures to multimodal systems were experimented [40].

It involves four layers of information representation in a multimodal perspective. The first layer (*Physical signals*) consists of signal-based descriptions of the input information detected by real or virtual sensors, such as sound frequency, energy and movement velocity. The second one (*Perceptual features*) comprises the extracted perceptual cues that are important for conveying expressive content, such as tempo, articulation, sensory dissonance and tonal tension. The third one (*Mid-level maps*) is obtained by segmenting and interpreting expressive gestures on a longer time scale and includes a gesture-based description of trajectories in spaces such as rhythm and harmony or fluency and impulsiveness in Laban’s Effort space. The fourth one (*Semantic layer*) consists of the high-level expressive content obtained from expressive gestures and can be organized as a conceptual, linguistic or symbolic structure. Emotions that can be described by a discrete label for basic emotions by the valence-arousal dimensional space or by other more specific dimensional models are an example. The more abstract the representation level, the less the information depends on the sensory modality from which it was derived, thus facilitating the integration of expressive communication between the different media.

This conceptual framework is highly appropriate for understanding and modeling the mechanisms underlying non-verbal communication through expressive gestures. It represents the information flow and layered mapping for expressive interaction in multimodal performance systems (see Figure 2), and it allows the development and design of effective models and algorithms for gesture processing in new expressive interfaces. Systems adopting this framework are being used in many applications and artistic scenarios. The previously seen *Voice painter* assumed this paradigm. The tools developed in MEGA by implementing this framework have been used for cross-domain mapping and expressive sound motion and for artistic works, such as *Medea* and *Casetta delle Immagini*, which we will see in Section 4.3 and Section 4.4.

##### Cross-Domain Mapping in Interactive Environments

Mapping expressive features from movement and audio to music and video was tested in several applications and compositions. The interactive game *Ghost in a Cave* was developed during the MEGA project to test the idea of non-verbal expressive communication as input in an interactive environment and as a basis for collaboration and involvement among participants [41]. Audio, motion and expressive cues are used as the game control. Two teams use either voice or body movement to navigate their avatar and compete through expressive gestures. As voice input, expressive and audio features from one player in each team, who can sing, talk or make any sound, are analyzed by the system. The expressive qualities of the body movements of the rest of the team are used to control the game behavior (Figure 16).

Performance theories and practices traditionally deal with the issue of orchestrating audience experience, engagement and reactions. Theater and games basically deal with planning emotions, forms of interaction, collaboration and relationships. This knowledge could contribute to design and dealing with human–machine interactions in general. Engagement is created through a balance between restrictions and possibilities. To orchestrate and make social and human–machine interaction possible in interactive environments, this balance is pivotal. The experiment showed that expressive gestures in music and dance and cross-domain mapping can be used as input control in an interactive game environment and as a basis for engagement and collaboration among participants using non-verbal communication.

##### Expressive Sound Motion

Digital audio technologies allow projecting and moving the virtual sound source in the immersive space of the listener. Composers’ curiosity about sound in all its dimensions, which was typical of the twentieth century, stimulated an interest in using these possibilities creatively. The use of space projection in an expressive way requires new paradigms for interaction and mapping. CSC investigated how sound movements could be considered a musical parameter and which expressive content can be conveyed to the listener. A model was developed that relates the basic components of sound movement (speed, articulation and path) to the intended conveyed emotions [42].

While the movements of sound in space have hitherto been used denotatively to convey explicit information about the source, this research opens up new avenues for communicating emotions by exploiting a channel so far used only intuitively by artists in a creative way. In the opera *Medea* (see Section 4.3), the model was used for cross-domain mapping of the expressive gestures of a player into expressive spatial sound movements.

#### 3.2.2. Sensory-Motor Expressiveness

Frequently in the music domain, expressive communication refers to emotional and affective content (for a detailed review, see [43]). We focused on a further kind of expressiveness content—*sensorimotor expressiveness*—which can be applied to both the compositional layer and the performance layer. Sensorimotor expressiveness refers to aspects not covered by musical and emotional expressiveness since it investigates the domain of cross-modal associations. The starting point is that metaphorical descriptions may offer possibilities to explain and understand aspects of the musical experience otherwise ineffable. Different perspectives may be suitable for different applications and contexts. Very often, verbal labels are used to describe the subjects’ experience when they listen to music. However, verbal descriptions only partially capture the musical experience. Non-verbal associations can be useful in providing the assessment of non-verbal experiences, too. Such an approach may help overcome some of the limits of communication necessarily imposed by language [44].

In order to understand sensorimotor expressiveness without using verbal labels, the associations between music and human movements were studied from the point of view of an action–reaction paradigm, being the interactions with other objects through contact, pressure and usually small amounts of motion. This kind of interaction can be simulated by haptic devices (i.e., devices that measure a user’s fingertip position, pressure and motion and exert a precisely controlled force vector response to the fingertip). Such devices enable users to interact with and feel a wide variety of virtual objects [45]. An action-based approach to a musical experience has the advantage of being applicable to different contexts and artistic forms based on gestural dynamics, such as painting or dancing. Such an approach can open the door to considering the visual arts of movement not only as they are implemented today in interactive art installations but also by realizing a strong closed-loop dynamic physical interaction.

During CSC studies on music performance, the concept of the performer’s expressive intentions was extended beyond emotions by including labels with sensorial connotations [46,47,48]. Cross-modal associations were investigated by using the semantic differential approach with non-verbal sensory scales taken from the visual, gustatory, haptic and tactile domains [49]. Sensory scales have been shown to provide a complementary understanding of the musical experience because they highlight aspects not easily accessible to natural language and offer new opportunities on which to base new interaction interfaces [50,51].

All these studies show that music experience consists of interacting qualities, aspects and factors and that cross-modal correspondences exist among a variety of sensory stimuli for music description. From a technological point of view, the understanding of metaphors to describe different aspects of music (affective, sensorial or physical) offers new opportunities on which to base new interaction interfaces. The relations between acoustic cues and metaphors could be used to develop systems for the automatic generation of metadata. Action-based metaphors could be used to enhance the interface of devices using gestural interaction with musical content, such as portable music players or musical video games [52]. Moreover, research on sensory scales can foster the development of innovative interfaces to browse audio digital collections. These new devices will allow users to interrelate in a spontaneous and even expressive way with interactive multimedia systems, relying on a set of advanced musical and gestural content processing tools adapted to the profiles of individual users, adopting descriptions of perceived qualities or making expressive movements [44].

### 3.3. Multimodal Interaction for Learning and Well-Being

In the 2010s, interaction studies opened new societal fields of research, such as inclusive learning systems for people with special needs using modeling of human motion tracking and non-verbal 3D sounds as a preferred communication channel.

#### 3.3.1. Multimodal Applications for Health and Care

The concurrent presence of multiple sensory channels in multimodal virtual environments allows users to dynamically switch between modalities during the interaction. In particular, sensory augmentation through additional modalities and sensory substitution techniques are compelling ingredients in presenting information non-visually when visual bandwidth is overloaded, when data are visually occluded or when the visual channel is not available to the user (e.g., for visually impaired people).

##### Integration of Haptics and Audition

Spatial perception and cognition rely on multimodal information. Understanding the combination and integration of different sensory modalities is an essential requirement for the development of multimodal human–computer interfaces. To assess the relative contributions of haptics and audition, a system for non-visually exploring virtual maps based on haptic and auditory feedback was developed. The height profile of a virtual map is haptically rendered through a tactile mouse. Global spatial information about the position within the map is provided by abstract anchor sounds located in specific points on the map and rendered in headphones by the 3D binaural system described above. The experiment showed that adding 3D auditory feedback significantly improves the amount of spatial knowledge acquired by subjects during exploration compared to unimodal haptic exploration [53].

The proposed concept, design and implementation allow effectively exploiting the complementary natures of the “proximal” haptic modality and the “distal” auditory modality. Multimodal systems for the representation of spatial information could largely benefit from audio engines that exploit known mechanisms of spatial hearing and technologies for 3D audio rendering.

In a related project, systems for technology-assisted motor rehabilitation (virtual rehabilitation) where interactive audio is used for enhancing motor learning in motor tasks were proposed [54]. The results support the assertion that sensory substitution systems have significant potential to improve functions for people with sensory impairment.

Further, if auditory signals can readily be used to drive motor adaptation, this suggests they may be a viable means to enhance sensory feedback for motor training applications, such as rehabilitation therapy and sports training. This study indicates the use of auditory feedback as an adjuvant for helping neurologic patients and other motor learners improve their movements.

##### A Vibrotactile System to Compensate the Acoustic Information to the Deaf Fencer

In combat sports such as fencing, the readiness of selecting visual, tactile and auditory information contributes decisively to the development of low reaction times, decision-making processes and winning strategies. In modern fencing, training and developing motor skills are mediated and constrained by the presence of the electronic system for detecting and signaling hits. The audiovisual display of the combat platform is greatly demanding for a deaf beginner fencer, both from a cognitive point of view and for the effectiveness of the athletes’ decision-making processes. An interactive system that reports hits and communications from the referee to deaf fencers was developed (Figure 17). The system includes a module with the function of interpreting the signals of the scoreboard and two vibrotactile wristbands worn on the unarmed athlete’s hand [55].

The vibrotactile interaction system has been shown not to require a cognitive overload and also tends to engender confidence in the user, causing an adjustment in their proprioceptive expectations of hitting.

#### 3.3.2. Large-Scale Responsive Environments for Learning and Inclusion

It is the authors’ opinion that inclusive learning for people with special needs is one of the most relevant issues to be addressed in the new millennium. As a matter of fact, since the 2010s, CSC has been exploring this field and developing interactive applications based on large-scale responsive environments and users’ engagement with expressive behavior using sensors such as 3D cameras, IR cameras and PhaseSpace Motion Capture.

Large-scale responsive environment applications have become a convincing leaning tool that could be used to playfully teach concepts of varying degrees of complexity. Full-body interaction (the user can walk inside an interactive space) is one of the most important aspects of the interaction design in large-scale responsive environments, as the user’s whole body acts as a living slider to control the application parameters. In doing so, the user’s movements can generate continuous data; they can change the sound frequency and amplitude according to position, for example, or they can make the user collide with interactive landmarks, causing the reproduction of predetermined audio and visual outputs. The process of interactively sonifying body movements has a deep emotional impact on users, as it fosters engagement, attention and curiosity. It is even more so when the whole body is involved and when many reality-based elements are concerned. As such, these environments encourage different learning styles and are particularly fit for the inclusive participation of people with disabilities and special needs, as they connect to the participants easily and on an emotional level.

Keeping that in mind, and using the *AILearn* responsive environment system, CSC created a series of serious games for music teaching and for teaching blind children how to walk: *Harmonic Walk*, for example, is an aid for teaching tonal harmony and melody harmonization; *Jazz Improvisation* is a tool for teaching music re-composition and score-part listening; *Following the Cuckoo Sound* is aimed at helping blind children to learn how to walk straight without walking sticks. The results, both qualitative and quantitative, showed a higher user engagement and a satisfactory number of successful attempts in performing formal tasks, pointing out the pedagogical value of using fun and competition in teaching [56,57]. Furthermore, this system has been fruitfully used to practice listening in English as a Second Language classes and to introduce interactive listening to children with severe disabilities [58].

These environments showed the great potential of interactive audio for the design of playful environments, exploiting the fun and the joy of discovery and forstering an enactive approach of teaching/learning.

### 3.4. Acoustic Analysis for Security

Using multimedia methodologies and technologies, CSC also carried out research works on environmental and workplace safety.

#### 3.4.1. Surveillance Systems

Unlike video cameras, audio sensors are omni-directional and do not need a direct line-of-sight with the signal source. For this reason, they can significantly improve surveillance systems based mainly on video technologies to complement vision with a sound-based localization of dangerous or interesting events in a monitored area. However, the use of audio-based surveillance systems is in its early stages. On one hand, sound-based surveillance can identify a sound source in a room and at a few meters distance using microphone arrays and signal processing techniques. The sounds captured by the microphones are evaluated through algorithms based on time delay of arrival estimation (TDOA) and delay-and-sum beamforming (SRP). On the other hand, audio localization techniques applied to large open areas (such as parks or squares) pose accuracy and precision problems not yet solved. In general, a microphone array error rate increases with the distance from the sound source and the angle between the source and the array position. Moreover, the localization accuracy depends on a number of interfering design factors, namely the array shape, its size, the distance between the microphones in it, the sampling frequency, the environment’s acoustic response and the presence of competing sound sources.

CSC developed a method to address the issues of multisource localization in large open spaces. Traditionally, the direction of arrival (DOA) is estimated by a network array system in the source and then processed by a Bayesian filter. However, in a noisy open space, acoustic sources are discontinuous and composed by several short-duration events. In this case, the filtering method is not the most effective way to track different sources. A more suitable method to assess DOAs relies on the incident signal power comparison (ISPC), as explained in [59,60,61].

The algorithm solves the ambiguous problem of correctly linking the DOAs from different arrays to the same source in a far-field condition with concurrent sources and can be a solution for multisource localization that requires a frame-to-frame analysis.

#### 3.4.2. Safety in the Workplace

A patent (WO2016120774) related to a method for the structural control of wooden poles was developed at CSC (CSC with OXYS SOLUTIONS S.r.l.; designated inventors: Alessio Biasutti, Umberto Bovo, Sergio Canazza, Maria Paola Gennovese and Antonio Rodà [62]. This invention (X-Poles, Figure 18) is fruitfully applied in controlling the structural strength (with particular reference to a possible presence of decay areas) of wooden poles adapted to support overhead lines of public/private utilities, such as power distribution or phones or similar networks. The following description will make explicit reference to such examples without losing the overall generality.

In general, a significant issue common to all public/private utility providers that normally use wooden poles for their distribution networks is to effectively identify the presence of decay areas in the wooden support structures. This is the main cause of structural strength loss in wooden poles, with a consequently considerable hazard, both during maintenance activities (which involve climbing poles and, in severe cases, may cause the poles to collapse and suddenly fall to the ground).

X-Poles overcomes the drawbacks and issues of the above-mentioned known method. In particular, X-Poles aims at providing a methodology for the structural control of wooden poles, making it possible to effectively and rapidly verify the presence of any decay areas or spots on the whole pole. X-Poles is substantially based on a special audio transducer that listens to the sound when the wood is hit, for example, with a sharp blow by a hammer or an equivalent tool. In the initialization phase, the sound captured by the audio transducer is appropriately digitized and sent to a computer. It is then broken down into its fundamental components in order to identify a specific feature set defining a “digital signature” of the obtained sound and along with it, the structure of the pole being hit. X-Poles therefore allows the simple and effective identification of any decay areas on the surface and in the internal part of the pole, particularly below the ground level and on the upper part of the pole (tip rot).

## 4. Sensors in New Music Research

In the following sections, some works from the 1980s to the 2010s are highlighted. They are noteworthy from a historical and scientific point of view. In these works, interaction plays a central role, involving researchers in the development of new musical ideas and simultaneously involving musicians as sources of creativity in scientific research.

### 4.1. Historical Background

The development of audio transducers such as the microphone and the loudspeaker have led to radical innovations in the musical world, first through the use of amplification—the transport of sound in places other than the location of the audio source—and then with the development of live electronics techniques. The first experiences of electronic music were born in the immediate post-WWII war period of the last century and mainly developed techniques for music composition in the laboratories using both real recorded sounds (*Musique Concrète*) and electronic synthesis sounds (*Electronic Music*) [7]. The main tool for composing sound and music was the magnetic recorder and concert performances that took place through the playback of music recorded on magnetic tapes. The first concerts with live electronics, therefore, dated back to the mid-1960s, when transistor technology allowed building electronic portable devices both for sound synthesis (synthesizer) and for the live processing of voices and/or instruments. In the 1970s, computer music began, but it remained in the laboratories due to the non-portability of the mainframe computers used in those years and the lack of possible gestural control of sounds. Musical compositions, therefore, continued to be played in concerts through magnetic tape recording, and the composers moved towards more abstract music conceived with formal structuralist techniques. Only in the 1980s was it possible to use the first computers equipped with audio processors and gesture control sensors in real-time.

Musically, CSC was born from previous experiences of the NPS Group in analogue electronic music and in live electronics of the ArkeSinth Group. In the 1970s, it became an important point of reference for the scientific and musical community engaged in the development of computer music [5]. In terms of control, a first attempt to overcome the limits of 1970s computer technologies was conducted by Teresa Rampazzi, who developed *With a light pen* (1978), composed on the computer with the Interactive Computer Music System (ICMS) by Tisato [63], which uses a light-sensitive computer input device on a CRT display to create the synthesis sounds and organize them in time.

With the advent of real-time audio processors, many composers who had given up using digital technology due to the difficulty of translating their musical ideas into algorithms and numerical data allowed themselves to be seduced by the new synthetic sounds that could be controlled live with new gestural freedoms. Additionally, the growth of new generations of musicians with computer skills who became assistants to the composers and performers of the live electronic parts of their compositions started [64].

That’s the case of composer Luigi Nono, who since the 1960s has made many electroacoustic music compositions by combining live voices and/or instruments with musical parts recorded and transformed by analogue techniques. Thanks to his skill, he was able to make the musical parts recorded on magnetic tape appear “live” when they were played in concert together with the live voices of the musicians. However, his dream was to play the electroacoustic parts of his compositions totally live. A dream that was realized in 1984 with the first performance of his opera *Prometeo*, totally live electronics, as will be described in Section 4.2.

New musical instruments also require new performers: with the emergence of live electronics [7] (chapter 8), now being used in a large music repertoire all over the world, a new professional figure with dual training, with both a musical and a scientific background, became necessary. Often, the music assistant is not just a performer but a collaborator in the composition of music and sound design [65].

At CSC, a close cooperation between the Electronic Music class at the Conservatory of Padova and the degree program in Computer Engineering at the University of Padova led to the successful training of many music engineers. This opportunity favored the emergence of a Venetian school of young composers who later established themselves in the world. From these origins, scientists, researchers and technicians continue to collaborate with artists using the new art–science interaction laboratory and CSC’s know-how as support for the innovation of expressive forms in music, musical theater and interactive multimedia arts, as demonstrated by the examples of musical works presented in the next paragraphs. Among these, *Medea* by Adriano Guarnieri in which the traditional means of musical theater (choir, voices, soloists and orchestra) and scenography are expanded with audio and video live electronics, strengthening the expressive force of each element, thus obtaining a choral stimulation of the public’s senses (Section 4.3).

The need to expand musical instruments beyond their traditional fields characterized a lot of musical literature of the second half of the 20th century, and this trend has further developed in the last few years of the last century using sensors, through which the gesture of the performer is related to the simultaneous electronic processing of the sound itself. A couple of significant examples of this musical research can be found in Section 4.5 with the works *Ogni emozione dell’aria* by Claudio Ambrosini and *Chemical free (?)* by Nicola Sani.

An opposite case of expansion of a musical instrument is offered by Stefano Gervasoni’s *Altra voce*, in which it is not the gestures of the performer that expand the sounds of a piano but the vibrations produced by actuators that transform the sound board of the instrument into a curious and fascinating “loudspeaker” (Section 4.6).

From instrument expansion to interactive music installation, the step is short, and this art form summarizes an artistic path that has found many achievements over the years, beginning at the turn of the new millennium. A significant example is Carlo De Pirro’s *Casetta delle immagini*, in which the public—in this case children—also becomes part of the show by interacting with sounds, images and colors activated by their gestures, as illustrated in Section 4.4.

### 4.2. Luigi Nono: Prometeo, Tragedia dell’Ascolto

A real leap in the quality of CSC music works was achieved in the early 1980s when a collaboration between IRCAM in Paris and the Venice Biennale, which started building the 4i System, a computer capable of generating sounds in real-time. The 4i system was immediately used by Luigi Nono as a musical instrument in the opera *Prometeo, tragedia dell’ascolto* (1984), which is definitely one of the most important musical works of the second half of the twentieth century, and technology plays a dominant role in it (Table 1). The composer renounced the traditional scenic elements and theatrical costumes of the opera to focus the audience’s attention purely on listening to the music in an acoustic environment specifically designed by architect Renzo Piano. The environment is made of a wooden structure hosting both the musicians and the audience for an immersive and enveloping experience while favoring the vertical dimension of listening. The music generated inside the structure naturally caused such a colossal soundboard to vibrate, along with the musicians and the audience, who would literally be incorporated in the resonant body. Thus, the wooden structure becomes part of the musical instrument (Figure 19, left).

At its first performance of the opera, which took place in the Church of San Lorenzo in Venice, the synthesis sounds of the 4i system were the first to invade the acoustic space of the church, emerging from the bottom of the wooden structure to spread in all directions, followed by the orchestral and chorus sounds, thus starting *Prometeo*. The 4i system was played completely live using a bank of six potentiometers and a dozen computer keys (Figure 19, right). Among the composer’s various requests, a gesture-controlled performance environment was created that could simulate a “coro velatissimo” (very veiled chorus) used in different sections of the opera. Therefore, the computer did not have to play a simple melody, but rather, it had to generate several sets of sounds that moved over time and could be aggregated around one or more musical pitches. Additionally, it had to simultaneously change the timbre both in a harmonic and inharmonic sense. The fluidity required by the transformations led us to prefer the potentiometer as a sensor for controlling the musical gestures and to choose the frequency modulation (FM) sound synthesis technique for high efficiency and versatility in timbre control [12].

#### Performance Environment and Mapping in *Prometeo*

The sound generation was realized by 24 FM voices with a sinusoidal carrier and modulator. The sound parameters of each voice were updated with a polling technique: each voice had a duration of about one second and repeated cyclically with a trapeziodal amplitude envelope. At the beginning of each cycle, the current values of potentiometers and control keys were assigned (see Table 2). In this way, the polyphony of the virtual choir evolved smoothly from one state to another in a very natural way. Potentiometer P1 controlled the number of active voices and therefore the choral density; potentiometer P2 controlled the global dynamic level; potentiometer P3 controlled the base frequency through which the carrier frequency and the modulator frequency of each of the 24 voices was calculated according to the carrier/modulator ratio set by potentiometer P4 and according to the harmonic structure key activated. For example, key K1 (unison) assigned the same base frequency to all 24 voices, while key K2 randomly assigned the base frequency to 12 voices and a semitone above (second minor) to the other 12, and so on until key K9, which evoked the A of the beginning of Mahler’s 1st Symphony, expanded 3–4 voices by 7 octaves each.

Potentiometer P5 controlled the tonal richness of each voice, passing from the sinusoidal sound (value 0) to a progressively increasing number of partials whose law depended on the value set by potentiometer P4. For example, when P4 = 1, the generated partial sounds followed the natural law of harmonics, whereas, when P4 = 0.5, only odd harmonics were obtained. When P4 was set to an irrational number, an inharmonic timbre was obtained. Potentiometer P6 (micro-intervals) controlled the “chordal mobility” of the sound, which was obtained by randomly varying the pitch of the sounds by a few Hz with respect to the base frequency. This effect was perceived more clearly with the 24 voices in unison and the sinusoidal timbre: slowly moving potentiometer P6 along its excursion, we moved from a single sinusoidal sound to a progressive increase in the beats among the 24 voices, which evolved from a fluid state to a state of tension that preluded dissonance.

The 4i system solo at the beginning of *Prometeo* could be described in its first moments: a sinusoidal chorus in unison (K1) with a slight mobility (P6) was subsequently brushed with a small timbral enrichment (P5) and then expanded to the tritone (K3) through small timbral and microtonal variations.

### 4.3. Adriano Guarnieri: Medea

In *Medea* (2002) by Adriano Guarnieri (Table 3), the musical protagonist is space since all the musicians are subjected to some form of spatialization, both physical and virtual, both static and dynamic. In this opera, the myth of Medea is represented by three female voices (soprano, “light” (i.e., non lyric) voice, contralto) and a single countertenor male voice embodying Jason, while all the other characters are represented by musical figures such as single soloists or groups of musicians. All performers are shot by microphones subjected to live electronic processing and diffused by a dozen loudspeakers mainly placed in front of and around the audience.

The sound movements in the PalaFenice theater, site of the opera’s premiere, took on a polyphonic nature in the virtual localizations of the sources and in the movement trajectories (Figure 20). Several soloists were placed in the hall. (For example, the bass flute player was placed behind the audience, although in some sections, s/he stubbornly moved between various speakers, having a wandering character.) On the other hand, two groups of four trumpets, placed opposite each other on the edge of the audience, produced sweeps of sounds that hovered over the heads of the audience, colliding and giving an extreme dynamic character to the Renaissance polyphony of the split choirs. In this specific case of spatialization, the motion sensors were the same recording microphones from which the amplitude envelope—used as a control signal for the virtual localization of the sound—was extracted through the following rule: the sounds of the horns with piano dynamics were localized in the physical position of the horns themselves; the sounds with plain dynamics were localized in the position of the opposing group of horns; and the virtual movement from one location to another took place over the audience through special sound shower diffusers hanging from the ceiling.

Equally fascinating was the spatialization of the choir, physically located at the back of the stage behind the orchestra. In different scenes, the chorus moved virtually to various positions with special acoustic effects, such as a surrounding chorus distributed in a circle around the audience, a very distant front choir, which appeared to be located outside the theater, and a so-called “celluloid” choir, which was placed in front of the audience on several levels as if it came from a projection on a screen of yesteryear with a celluloid grainy film simulated electroacoustically through micro-spatial movements of the sound of each chorister on the two-dimensional space of the proscenium. The orchestra was also subject to spatialization interventions. On the general level, it happened through the transparent amplification technique, and on the individual level, the movements of some percussion instruments were relevant. These included two metal cables played by two performers whose sonic gesture of violent tearing of the cable was transformed into an acoustic meteor launched into space in the longitudinal front-rear direction. However, perhaps the most interesting spatialization concerned the four trombones, which the composer had chosen to place separately throughout the audience, thus offering a natural spatialization to their sounds. Moreover, the sound of the trombone had been the principal object of specifically carried out research to make the virtual spatialization loaded with expressive meanings consistent with the gestures of the performer himself (Figure 20). Such a feature was implemented in the musical passages described in the score, with the technique illustrated in Section 3.2.1.

### 4.4. Carlo De Pirro: Casetta Delle Immagini

The *Casetta delle immagini* (Little House of Images) (Table 4) is an example of an interactive multimedia artistic installation by Carlo De Pirro and part of a larger context, namely that of *Pinocchio’s Square*, a work created on the occasion of the World Expo held in Neuchâtel, Switzerland, between May and October 2002. In the previous installation *Caos delle sfere* [5], another interesting work realized by De Pirro at CSC, an electromechanical pinball was used as input sensor.

Organizers placed several wooden platforms (80×30 m), and each platform was dedicated to an artistic theme or a scientific discipline. *Pinocchio’s Square* was installed on the platform dedicated to Artificial Intelligence and Robotics (Figure 21).

The work was part of the MEGA research project, funded at that time by the European Union, with the involvement of several universities (including the Padova University) and telecommunication companies and European multimedia producers. MEGA was focused on the modeling and communication of expressive and emotional content in non-verbal interaction by means of multisensory interfaces in mixed reality environments. Starting from the idea that music and movement are privileged means to communicate emotional content, the project systems were developed to analyze the expressive content of movements and to control the interaction with audio and video materials. The project led to the realization of a software environment (EyesWeb), which allowed the analysis and control of multimedia objects. Altogether, there were four types of sound inventions in *Pinocchio’s Square*:Sound of a Disklavier transformed into a light wind harp: the sounds of the piano sent vibration through thin copper strings, creating wings of butterflies attached to a rough body of colored glass;Synthesis sounds and sampled sounds: a metamorphosis from water to thunderstorm, to voice, to lightning, to fire, to whispers and to lions was performed;*Carillon of matter*: 30 electro-magnets on a rotating board to strike glass sheets, harmonic steel sheets and rotating saws;*Casetta delle immagini*: described below.

***Casetta delle immagini.*** It was a magic room designed for children in which every gesture became a sound, image or color (cross-modal interaction). Visitors were involved in communication between expressions and emotions through a non-verbal interaction implemented through multisensory interfaces within a mixed reality environment. The original work involved filming the gestures of the visitors and extracting high-level features from them by a system of two cameras and then projecting the acquired and appropriately processed images (through mapping strategies) on two video projectors. The images were also associated to some rhythmic-musical sequences diffused on a four-loudspeaker system. The process of the filmic sequences was controlled by associating the gestural movements to the rhythmic-musical sequences. In this way, the users could interact through their gestures with visual and sound objects, which were going to characterize the surrounding environment.

The process carried out on the input images included:Extraction of the user profile;Profile manipulation (stretching, rotation, color change, diffusion, etc.);Projection of the profile in virtual environments;Distortion effects of the environment and/or the user (fisheye, vortex and others);Color processing (subtraction, substitution and addition);Editing of visual objects (select areas, cut, copy and move).

The equipment used for the realization of the House of Images included:Three Personal Computers (V1, V2, A1) equipped with:−O.S. MS Windows 2000;−Pentium III 800 MHz CPU with 128 MB of Ram;−Quantum Hard Disk 16 GB;−CDRom Reader;−Dual-monitor video card with video capture (Matrix 32 MB);−Sound card not integrated into the MainBoard (Sound Blaster Live 1024);−EyesWeb software version 2.4.1;Two video cameras with adjustable lenses ≥ 28 mm;Four video projectors with SVGA input (Epson EMP50) (Projector field at 1 m diagonal of 71 cm to 11 m diagonal of 7.6 m; 35 mm 63 degrees; 50 mm 47 degrees);Amplification: four-channel mixer + four amplified audio monitors.

The team who developed the work also set a number of goals to be achieved through their work:Using Eyesweb patches dedicated to motion analysis for high-level features measurement;Studying different mapping strategies between motion and sound;Studying the effect of these mappings on an audience of children;Testing the reliability of MEGASE in a long performance (18 h/day for 6 months) subjected to a huge audience (4 million visitors).

**Structure of the work.** Within the EyesWeb environment, each file is called a patch. In the original work, the House of Images was composed of three patches: two were dedicated to the processing of movement (input) and images (output), and another delegated to the processing of audio sequences (output).

The processing of the output images was subdivided into seven distinct and independent parts to realize different effects that the user could observe, namely: Cage, Chaos, Ghost, Monet, Profile, Super Chaos and Worms. The Ghost’s effect can be watched at: https://youtu.be/HQlki4V7Dh8 (accessed on 15 March 2022).

The two patches dedicated to video processing were then connected to the third patch. As such, it was possible to realize cross-modal interaction and therefore manage the sound output through visual input (the volume of different sounds depended on users’ movements) (Figure 22). Originally, patch 1 was installed on V1 and was dedicated to the processing of the video captured by the first camera; patch 1 was divided into two parts: analysis of the video stream for the calculation of high-level features and effect synchronization. It was connected via MIDI to V2 and A1. Patch 2 was installed on V2 and was dedicated to the processing of the video captured by the second camera; based on the synchronization controlled by patch 1, the video stream was processed, and the results were displayed on the projector. V2 was connected via MIDI to A1. Finally, the last patch was installed on A1 and was dedicated to rendering the audio content based on the features received from patches 1 and 2. The output was then directed to the speakers.

### 4.5. Ambrosini: Ogni Emozione Dell’Aria

*Ogni emozione dell’aria* (2011) (Table 5) is a work for bass clarinet and live electronics composed by Claudio Ambrosini, in which the translation of extra-musical gestures into sounds is used to increase the polyphony of the piece. In this work, both hands of the bass clarinet performer are tracked by a real-time motion capture system in order to directly control the live electronic system for sound processing [66]. The Impulse Phasespace system was used with two position sensors placed, respectively, on the right and left wrist of the clarinetist. The score calls for specific movements of the player (i.e., opening arms), and the movement data captured by the system are used to map the position of sound in space and to add expressive intentions and new layers to the composition.

In this work, each hand is seen as a single independent body. For example, in some sections of the piece, the left hand controls the location and movement of sound in space, while the right one is connected to timbral effects (i.e., harmonizing, non–linear distortion, etc.). In other sections, however, the gesture-processing association varies according to the use of both hands in the performance of the clarinet parts and the desired type of sound processing. The performer’s gestures are notated precisely in the score in order to replicate performances in a deterministic way. At the same time, these new compositional parameters (gesture movements) preserve the natural inclination of musical expressiveness to be adapted to individual performance aesthetics (which is generally called musical interpretation). For example, Figure 23 shows a passage where spectral sound processing and transposition effects are applied, which, in the composer’s words, are described as, “The right hand seems to help the sound of the clarinet to come out and then back in again”. In the top right, the movement of the right hand to obtain these effects is indicated.

Figure 24 shows another example in which the clarinetist moves both hands to control the sound of a long note, while the instrument is supported by the knees. In this case, the right hand controls a grain and ring modulation effect, adding more synthetic voices to the actual sound of the instrument. In the spatialization, the gesture ideally pushes the sound behind the audience, but then, the sound slowly returns to its initial position, as if attached to a rubber band, unless there were new upcoming sounds creating new tension sending it away again. In this case, the sound processing enables the composer to add a metaphorical and dramaturgical layer through the movements of the hands, representing the wings of a flying bird, even when they control the processing of the sound (Figure 24, right).

While the type of gesture and evocative meaning are indicated in the score, how to proceed to achieve these effects is not written down. This is left to the taste and skill of the performer, but, as we can well understand, this lack of notation poses serious problems for the long-term preservation of the composer’s ideas and for performances in the more or less near future.

The use of precise position sensors is particularly interesting in musical productions with live electronics due to the wide versatility of application and the simplicity of the sensed parameters (*X*, *Y*, *Z*). It guarantees their long-term sustainability, an important factor for the music’s survival. For example, detecting the position of the violinist’s wrist allows tracing the movement of the bow and, from this, to implement different types of processing. This is what has been achieved in *… fili bianco velati …* (2010) by Adriano Guarnieri for violin and live electronics and in *Abitata dal grido* (2011) for cello and live electronics by the same author.

In *Chemical free (?)* (2014) by Nicola Sani, the position sensors are applied to the wrist of the pianist and of the hyperbass flutist (Figure 25, rleft). This instrument, designed by flutist Roberto Fabbriciani, is particularly suitable for live electronic processing as the performer’s hands are free, and therefore, they become ideal controllers for the augmented instrument (Figure 25, right). Another interesting application of position sensors is related to the performance of acousmatic pieces that can become spectacular with the presence of a live performer. An example that has been met with considerable public interest is the performance of György Ligeti’s *Artikulation*, in which the performer is free to move around the stage controlling the spatialization and dynamics of the acousmatic fixed media through body movements and gestures.

### 4.6. Gervasoni: Altra Voce

*Altra voce* (2015–2017) by Stefano Gervasoni (Table 6) is a musical composition for piano in which the actuators have a prominent, almost magical role in the piano repertoire. It is well known that Robert Schumann suffered from auditory hallucinations and that they influenced him in his musical production. Gervasoni takes inspiration from some of Schumann’s works to create five short movements dedicated to the German composer: the first of these comes from *Zwölf Vierhändige Clavierstücke für kleine und große Kinder op. 85 n. 12*, in which the author writes in the score an aria that should not be played by the pianist but only sung internally to give expressive strength to the real musical part played. In *Altra voce*, Gervasoni chooses to give sound to this internal air by making it magically appear like a siren song from the piano itself. In reality, it is an electroacoustic sound emitted by six vibration actuators that excite the soundboard of the piano. In the subsequent movements of the composition, again through actuators, the piano sings with the real voice of a soprano or is tinged with electronic sounds in dialogue with the pianist’s live performance.

## 5. Conclusions and Perspectives

With the advent of digital technologies, computers have become generalized tools for producing music: if music is the message, a computer is the instrument that conveys that message. Music, then, can be seen as a creative form of human–human communication via computers, and therefore, research on human–computer and computer–human interfaces becomes important. Research on sensors (for the acquisition of signals from the physical world for input to computers) and actuators (as rendering devices of media signals to the public) is essential for expressive artistic communication.

Since the early 1970s, a group of researchers and musicians has been working at the Centro di Sonologia Computazionale (CSC) of the Padova University on music technology and computer music. In this paper, the authors have reviewed CSC research in the field of music technologies while focusing on the aspects of interaction between musician and computer and between computer and audience. In particular, we discussed input devices for detecting information from gestures or audio signals and rendering systems for audience and user engagement. Moreover, specific attention was devoted to the so-called mapping—i.e., the link strategies between the outputs of a gestural controller and the input parameters of a signal processing unit. To take into account the implicit information conveying the expressive content, which is very important in art and music, CSC developed a multilevel conceptual framework, which allows processing and coordinating multimodal expressiveness. Some paradigmatic musical works were presented in detail and have started new lines of both musical and scientific research. In these works, from the 1980s up to now, interaction has played a central role, involving researchers in the development of new musical ideas and involving musicians as sources of creativity in scientific research at the same time.

The use of information technology in art gave rise to new challenges for the preservation of cultural heritage. If technology has been a stimulus to new forms of artistic creation, it has simultaneously become the cause of their rapid deterioration and their shorter life expectancy. Their great dependence on technology makes them particularly vulnerable, and there is a serious risk of losing an important part of today’s culture. Due to its immaterial nature, music was one of the earliest types of art to explore the creative use of new technologies. The preservation of such technology-enhanced artistic creations presents very different problems from those posed by traditional artworks. To safeguard this heritage, digitizing the content of recordings and documents is not enough, and the possibility of a faithful performance in a long-term perspective should be assured. Sensors and interaction technology play central roles in this. CSC is actively engaged in proposing new paradigms for the preservation of digital art, and such strategies can be applied both to the devices and to the behaviors derived from the software and its interaction with the user [67,68,69].

The human voice was the first musical instrument and is still the main instrument in most musical genres. The expressive power of the voice arises because it is an instrument inside the human body. Therefore, musical control takes place in a direct and unmediated way, as happens in musical instruments that act as “mechanical extensions” of the human body. The development of neuroscience and biomedical engineering has brought new insights into the human body by developing sensors capable of picking up “intimate” signals that can eventually become sophisticated controllers of musical gestures. The timbre of sounds was the most-investigated musical parameter by 20th century composers, and parallel scientific research has identified its multidimensional characteristics. The idea of developing a performance system able to directly connect appropriate intimate signals of the human body to the sound synthesis, allowing the music imagined to be played in real-time, has therefore become fascinating.

On the output side, the paradigm of an enactive interface can be extended to other art forms, such as time-based visual arts (e.g., movies, animation, etc.) and choreography (e.g., dance, theater), where at any stage of the production process, sensory artistic events are evolving events. The concept of MPS can be generalized as a physical mediator that is able to produce sensory stimuli through a human body exerting an action on it. All the arts that need such a mediator are necessarily time- and interaction-based. The system becomes an interactive instrument that allows the design of ”objects” that can be manipulated to produce moving events. In this way, it is possible to trigger true integration between the dynamic arts [20].

The issue of mapping between sensor output and signal processor input could benefit from a paradigm shift. The multilevel model presented allows a mapping in which explicit knowledge is best exploited and formalized at the most appropriate level of abstraction. It adopts a symbolic paradigm of information processing and allows better understanding and designing relationships according to desires. However, this works when the amount of input and output is reasonably small, and the knowledge of these possible relationships is fairly well developed. When massive data can be collected, a data-driven approach to AI, on the other hand, can make it possible to establish direct relationships between many more parameters, which would be more effective in practice, even if not easily interpretable intuitively.

In music, AI has long been used effectively for automatic composition [70] and for the expressive rendering of musical scores [71] but relatively little for parameter mapping in interactive performative systems. As we mentioned before, the musician has to study his musical instrument for many years to adapt to its physical characteristics and make it become an extension of his body. Today, electronic instruments, as is well known, evolve so rapidly that it is impossible to apply the traditional study technique, and therefore, it is advantageous to overturn roles by exploiting AI techniques. In other words, instead of being the musician who “learns” the instrument, it is the instrument that, thanks to AI, “learns” the performer’s “intimate” gestures and translates them into sound. In this way, different types of electronic instruments could be “played” starting from a generalized gestural language, possibly expandable according to the peculiarities of the single instrument. This is a line of research that is worth pursuing in the future.

CSC has pursued an approach in which the research methods of art, science and technology converge in common principles while also valuing the multiplicity and variety of knowledge.

In adopting such a perspective, the combination of artistic and scientific research practices implies that actions in one domain affect actions in the other and vice versa. Such an approach is required by current developments in culture and technology and is particularly beneficial for research on digitally mediated human–computer and human–human interaction. Interactive art technology is a very suitable domain to experiment such integration. Within this framework, the knowledge of culture in the humanities, beyond being reflective and critical, may have the inherent power to contribute to innovative techno-cultural developments [72].

Moreover, this kind of research will increase the opportunities for artists, scientists, industries, cultural intermediaries and the public in new emerging forms of technology and creativity [73]. In artistic performance, interactive technology can be linked with human-centered approaches, allowing the user to adjust, control or generate content. Interactive technology thereby encourages the application of expressive interactions elicited by new emerging forms of technology.

## Figures and Tables

**Figure 1 sensors-22-03465-f001:**
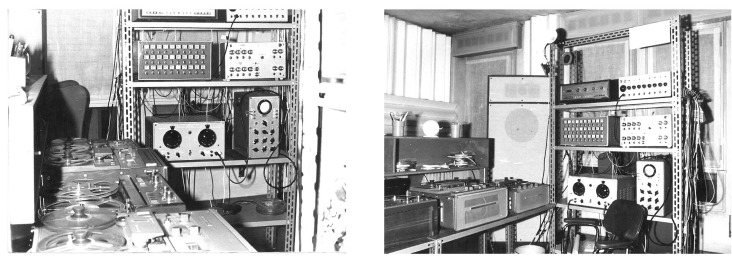
The analog studio *Nuove Proposte Sonore* (NPS) for electroacoustic-music in Padova, where some components of CSC initiated their activity.

**Figure 2 sensors-22-03465-f002:**
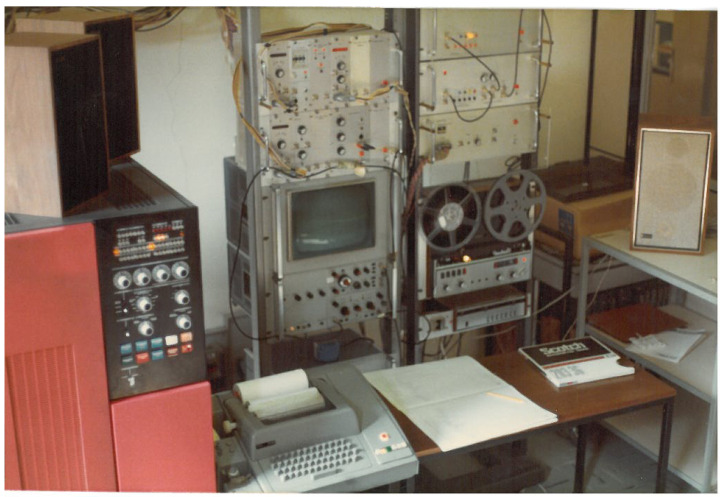
Hardware and software systems used by CSC in 1979 for recording, sound synthesis and music processing.

**Figure 3 sensors-22-03465-f003:**
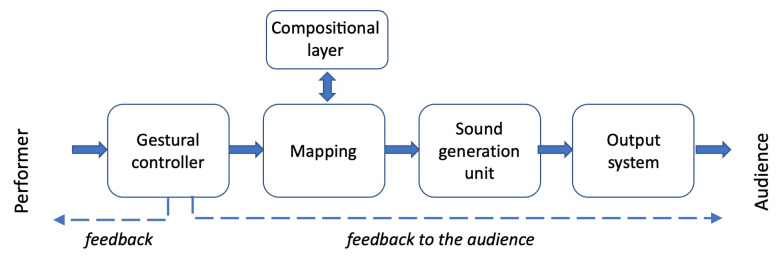
Scheme of a Music Performance System.

**Figure 4 sensors-22-03465-f004:**
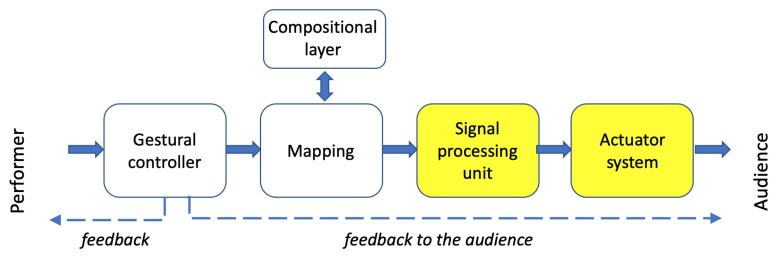
Scheme of a Multimodal Performance System. In yellow the units dealing with multimodal signals.

**Figure 5 sensors-22-03465-f005:**
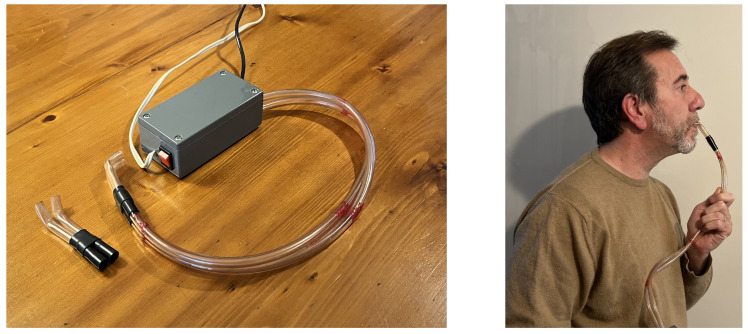
The interface designed for detecting postures of the oral cavity. It has a small loudspeaker, a microphone and two physical waveguides: one feeds white noise inside the mouth, and the other picks up the signal filtered by the oral cavity.

**Figure 6 sensors-22-03465-f006:**
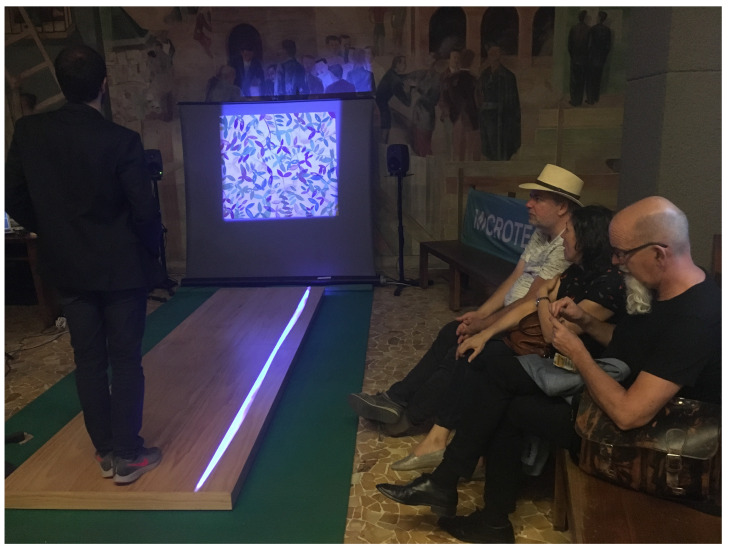
The interactive sonorization of a painting of Hartwig Thaler (first person on the right) at the European Researchers Night (2018).

**Figure 7 sensors-22-03465-f007:**
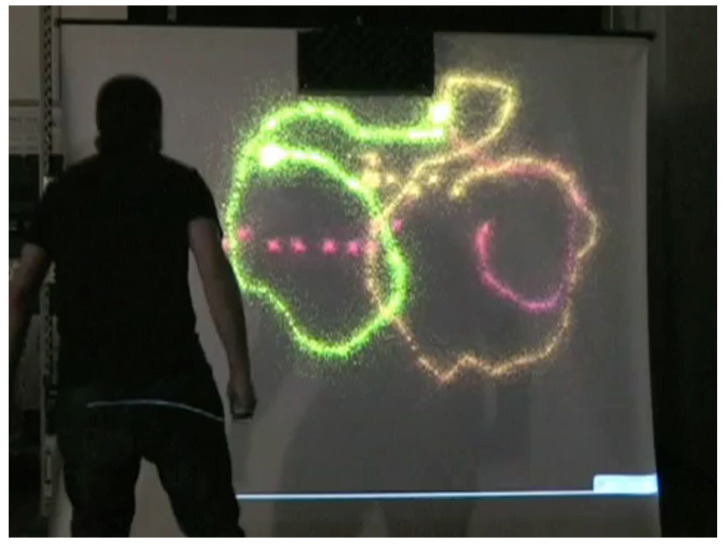
The installation *Voice painter* using body movements and vocal expression to create a painting.

**Figure 8 sensors-22-03465-f008:**
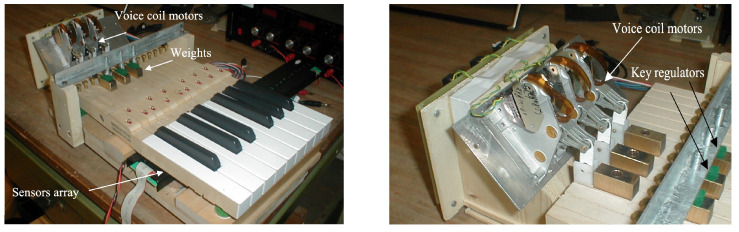
The Multi-Instrument active KEYboard (MIKEY) with a force feedback to the touch.

**Figure 9 sensors-22-03465-f009:**
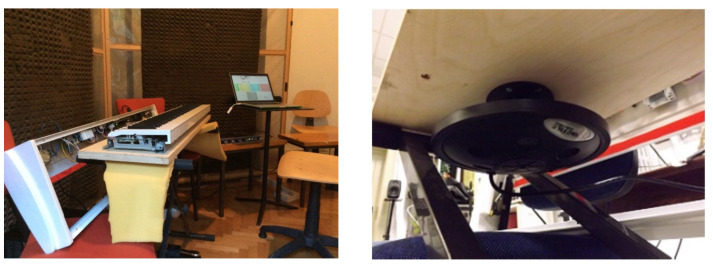
Experimental setup for studying vibrotactile feedback on digital piano (**right**). One of the transducers used to convey vibration at the keyboard (**left**).

**Figure 10 sensors-22-03465-f010:**
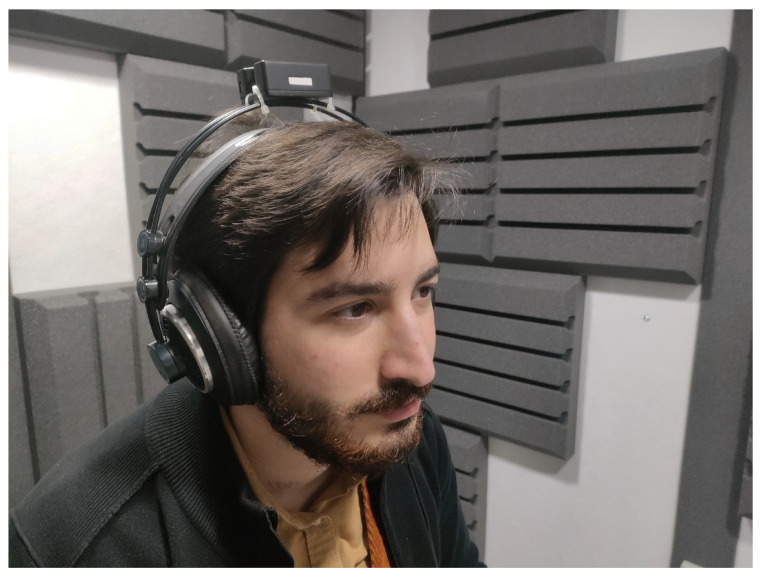
Motion-tracking headphones. By controlling the individual HRTF structural model, it allows perceiving the position of the virtual sound source independent of head movements.

**Figure 11 sensors-22-03465-f011:**
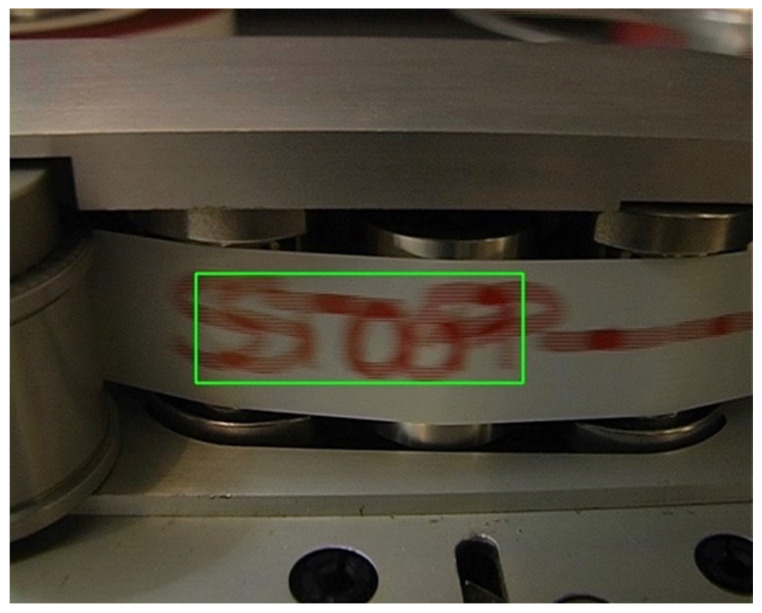
Example of words written on the back of a tape.

**Figure 12 sensors-22-03465-f012:**
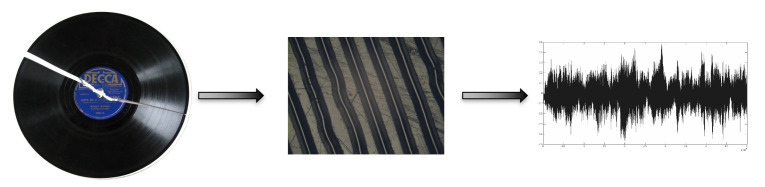
Photos of the GHOSTS system for reconstructing the audio signal from a still image of the surface of shellac discs.

**Figure 13 sensors-22-03465-f013:**
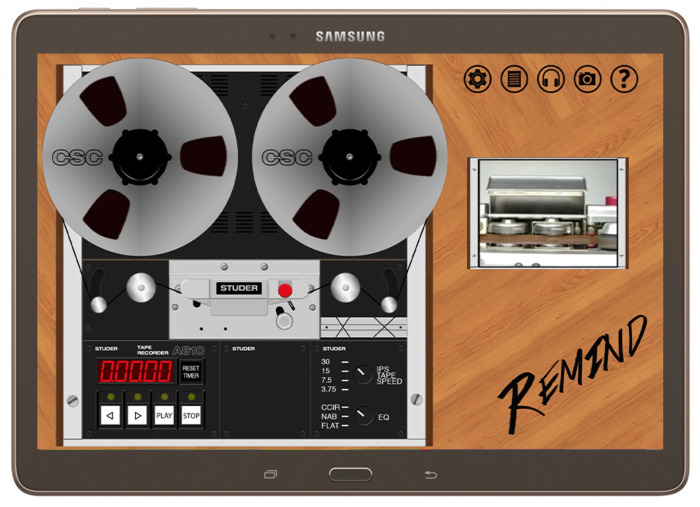
User interface of the REMIND app. A video of the original tape is shown separated from the tape recorder to improve readability.

**Figure 14 sensors-22-03465-f014:**
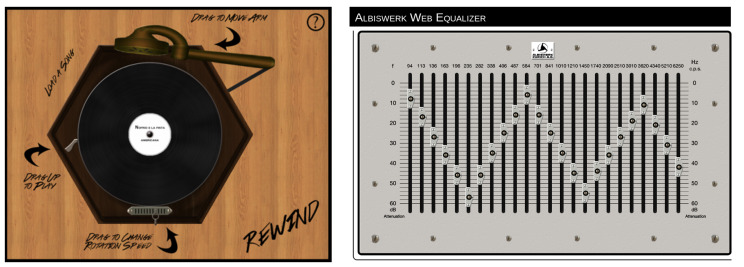
Virtualization of an old gramophone (**left**). Virtualization of Albiswerk model 502/50 equalizer (**right**): a library of the historical equalization curves used in electroacoustic music and shellac phonographic discs is included.

**Figure 15 sensors-22-03465-f015:**
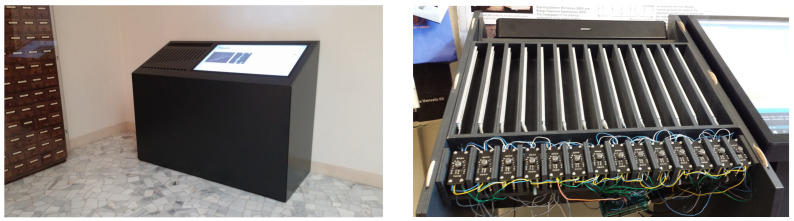
Interactive installation to enhance the viewing experience of the antique Pan flute at the Museum of Archaeological Sciences and Art (MSA) at Padova University: installation in the venue (**left**); details of the blow flute section (**right**).

**Figure 16 sensors-22-03465-f016:**
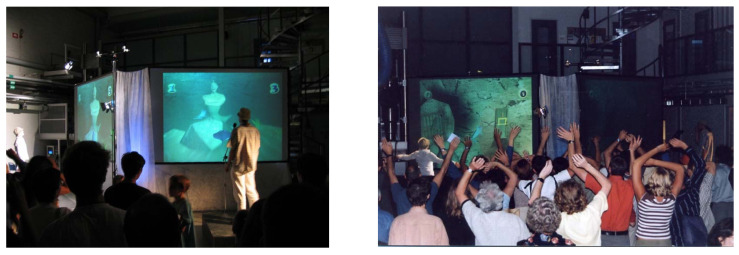
*Ghost in a cave*: The players’ avatars have entered the “happy” cave. The motion input player on the left and the voice input player on the right (**left** image). Team 1 is helping the motion player to reach the gate of the cave displayed on the screen to the left (**right** image).

**Figure 17 sensors-22-03465-f017:**
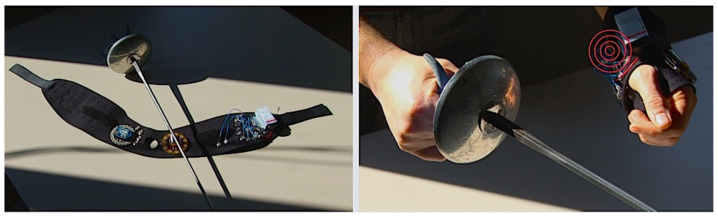
Glove with wrist strap for radio communication from the fencer to the scorekeeper and for actuator control.

**Figure 18 sensors-22-03465-f018:**
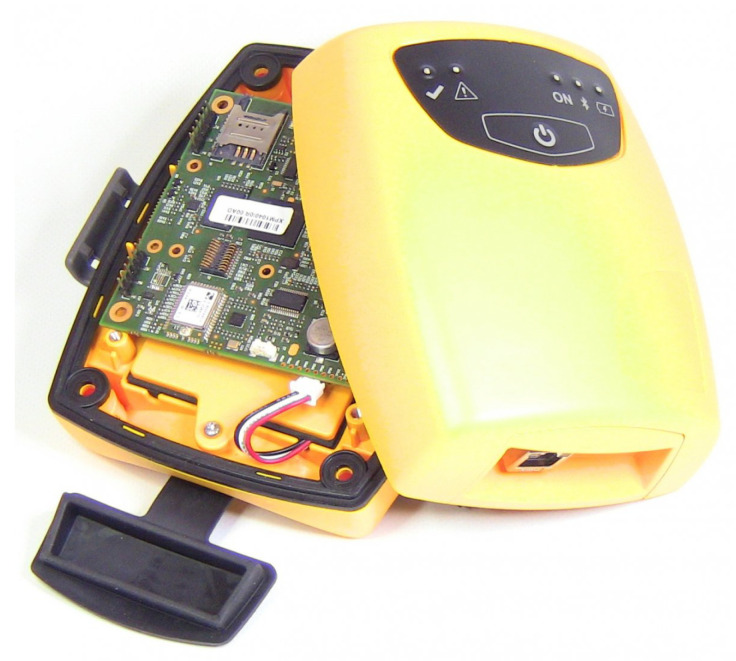
X-Poles is fixed to the pole through an elastic strap. The test starts with a vibration measurement generated by a hammer. Using a built-in accelerometer, the device collects the pole vibrations and then extracts the natural eigenfrequency and evaluates the Ultimate Breaking Strength of the pole.

**Figure 19 sensors-22-03465-f019:**
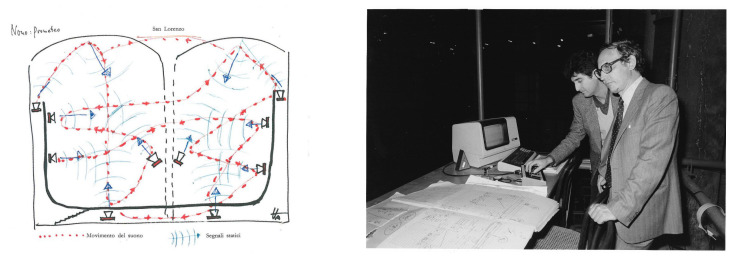
*Prometeo, tragedia dell’ascolto* by Luigi Nono. Sound movements inside the wooden resonating structure: as drawn by Hans Peter Haller, sound director (**left**). Alvise Vidolin (live electronics performer) and Giuseppe Di Giugno (designer of 4i System) during the rehearsal in Venice in 1984 (**right**).

**Figure 20 sensors-22-03465-f020:**
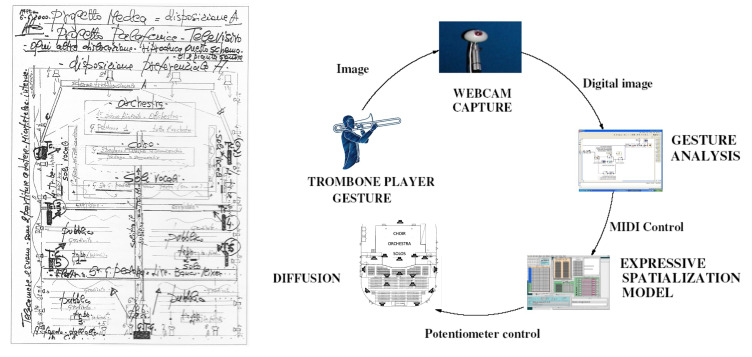
Guarnieri’s manuscript (first draft) with the arrangement of musicians and electronics at the PalaFenice (**left**). Diagram of the information flow from the player’s gesture to the expressive spatial diffusion of the sound (**right**).

**Figure 21 sensors-22-03465-f021:**
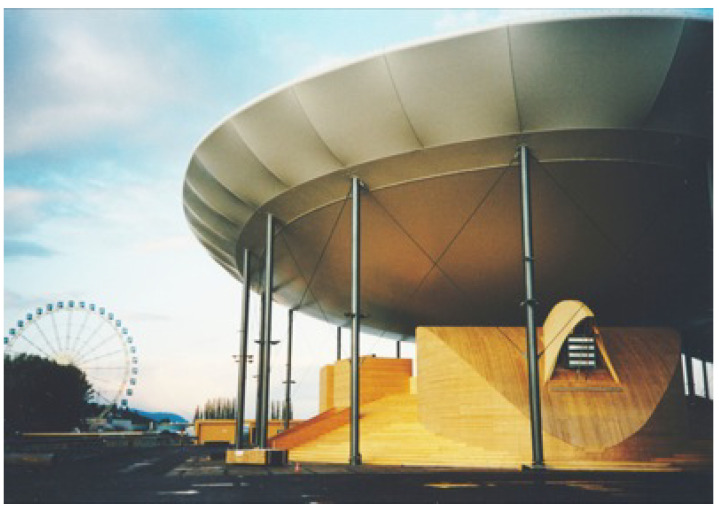
Interactive multimedia installation, *Pinocchio’s Square*, exhibited at the Expo in 2002 in Neuchâtel, Switzerland, from May to October 2002. Rear view of the massive wooden structure that housed the installations inspired by Pinocchio’s story.

**Figure 22 sensors-22-03465-f022:**
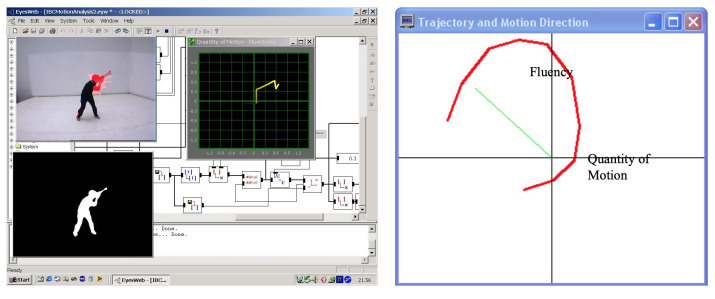
Interactive multimedia installation *Pinocchio’s Square*, exhibited at the Expo 2002 in Neuchâtel, Switzerland, from May to October 2002. One of the EyesWeb patches used during the exhibition, which maps expressive gestures into the Fluency/Quantity-of-Motion mid-level space.

**Figure 23 sensors-22-03465-f023:**
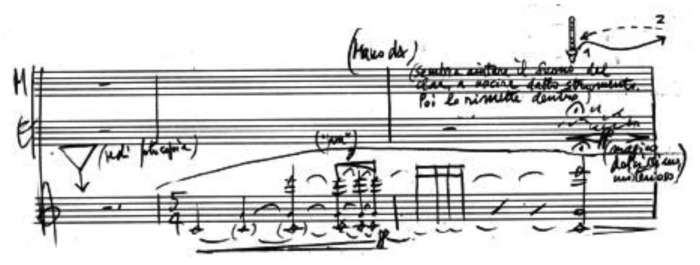
Excerpt of the score of *Ogni emozione dell’aria* by Claudio Ambrosini. Right hand movements are indicated in the top right.

**Figure 24 sensors-22-03465-f024:**
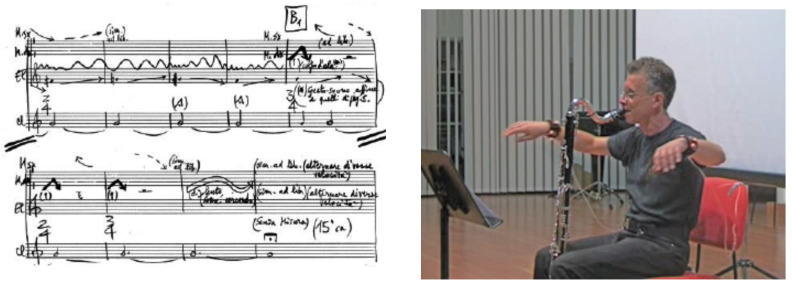
Gestural control of Ogni emozione dell’aria by Claudio Ambrosini: excerpt of the score with right hand movements (**left**); gestural performance by Davide Teodoro (**right**).

**Figure 25 sensors-22-03465-f025:**
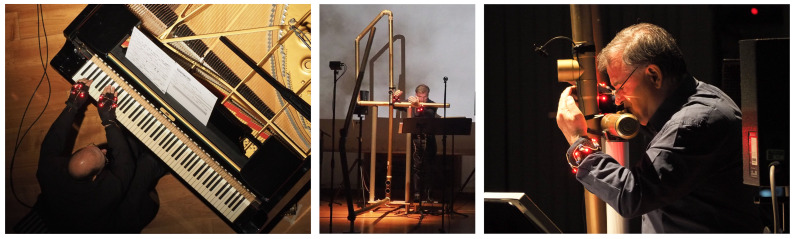
*Chemical free (?)* by Nicola Sani. Position sensors are applied to the wrist of the pianist (Aldo Orvieto) and the hyperbass flutist (Roberto Fabbriciani) to control the performance system.

**Table 1 sensors-22-03465-t001:** Luigi Nono: *Prometeo, tragedia dell’ascolto*.

*Prometeo, la tragedia dell’ascolto* (1984) music by Luigi Nono, texts by Massimo Cacciari, for vocal
and instrumental soloists, choir, orchestra, live electronics, and the 4i system was commissioned
by the Venice Biennale and the Teatro alla Scala in Milan.
Computer production was by Sylviane Sapir, Alvise Vidolin and Mauro Graziani, and the work
had its premiere on 25 September 1984 in Venice, in the Church of San Lorenzo.
Score 1337876 Ricordi, Milan, 1985. CD: EMI Classics 7243-5-55209-2-0, 2, London, 1995;
Stradivarius STR 37096, Milan, 2018.

**Table 2 sensors-22-03465-t002:** Control parameters of *Prometeo*: potentiometers are used to input continuous values, and keys are used to select the harmonic structure.

Potentiometers		Keys	
P1	Polyphony	K1	Unison
P2	Dynamic	K2	Ascending second minor
P3	Base Frequency	K3	Tritone
P4	Carrier to modulator ratio	K4	Fifth
P5	Modulator peak deviation	K5	Descending second minor
P6	Microintervals	K6	Descending fourth
		K7	Eighth
		K8	Fifth and octave
		K9	Seven octaves

**Table 3 sensors-22-03465-t003:** Adriano Guarnieri’s *Medea*.

*Medea, video-opera in three parts freely inspired by Euripides* by Adriano Guarnieri, 2002. For video
sequences, solos, choir, orchestra and live electronics. Commission of Teatro La Fenice. First
performance: PalaFenice, Venice, 18 October 2002.
Computer production: Bernardini and Vidolin. Score: Ricordi, Milan, 2002.

**Table 4 sensors-22-03465-t004:** Carlo De Pirro: *Casetta delle immagini*.

*Casetta delle immagini* (2002) interactive multimedia artistic installation by Carlo De Pirro was
created on the occasion of the 2002 World Expo held in Neuchâtel, Switzerland, between May
and October 2002.
Carlo De Pirro, composer; Roberto Masiero and Sergio Camin, scenography; Gustavo Groisman
and Luca Missio, architects; Sergio Canazza and Antonio Rodà software developers.

**Table 5 sensors-22-03465-t005:** Claudio Ambrosini: *Ogni emozione dell’aria*.

Claudio Ambrosini: *Ogni emozione dell’aria* (2011), for bass clarinet and live electronics. Davide
Teodoro bass clarinet; Amalia de Götzen, live electronics and motion capture; Alvise Vidolin,
sound director. First performance: SMC-2011, Padova.

**Table 6 sensors-22-03465-t006:** Stefano Gervasoni: *Altra voce*.

Stefano Gervasoni: *Altra voce—Omaggio a Robert Schumann* (2015–2017) for piano and transparent
electronic devices. Edizioni Suvini Zerboni S15505Z. CD: Kairos 0015082KAI, Vienna 2020. Aldo
Orvieto, piano; Alvise Vidolin, live electronics and sound direction; Monica Bacelli, mezzo-soprano.

## Data Availability

Not applicable.

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
