# Peer review of "Gesture, Music and Computer: The Centro di Sonologia Computazionale at Padova University, a 50-Year History"

_sensors, 2022, doi:10.3390/s22093465_

Round 1

Reviewer 1 Report

This paper is very interesting. I am touched by these paradigmatic musical works. Due to possibilities of rapid deterioration of music technologies, to share and record of today culture and music is important point.

Author Response

Thanks

Reviewer 2 Report

In this paper the authors present a review of the research in the field of music technologies at Padova University by the Centro di Sonologia Computazionale, focusing on scientific, technological, and musical aspects of interaction between musician and computer and between computer and audience.

The work presented is interesting and its content fits the topics of the Special Issue “800 Years of Research at Padova University”.  The authors present a review of research in the field of music technologies carried out at the CSC, focusing on the aspects of interaction between musician and computer and between computer and audience. This Special Issue is very specific. However, in this context it is an interesting work.

The described research works are interesting

Figures are adequate.

References are adequate in number.

However, some aspects could be analyzed in more detail to improve the quality of the article.

A short description of the most relevant contributions and findings of each research project described would be interesting.

It is important to know the work carried out in the CSC but it is also important to know its framework in a more global context. Thus, despite the relevance of the work and research carried out at the CSC, it would be interesting to include a framework of the works presented within the scope of what was being done internationally in each of these areas. This contextualization would be important to help understand the relevance and the innovative ideas of the works that were developed and that are described here. This would make the article more interesting to a wider audience.

It is a very descriptive article.

The CSC’s visions of the future, presented in the last section, are quite generic.

There are also some parts of the document that need to be revised:

(195-198) list numbering should be revised

(References section) There is a typo in reference 38 "20104"

Author Response

We thank the reviewers for their comments and for understanding the purpose of the article. The article was written for a special issue “800 Years of Research at Padova University” and it aims to present the research of Centro di Sonologia Computazionale on scientific, technological and musical research for music interaction.
We resubmitted the revised version, indicating in blue the new parts
We thank reviewer 2 for making very pertinent suggestions on how to improve the quality of the article and for encouraging us to contextualize our research. To this purpose, we inserted some paragraphs for each research topic. Moreover, we added 15 more references for a better understanding of the context.
We extended the Conclusion section with a few more considerations.
We corrected few typos, as pointed up by the reviewers.
Figures 20, 22 have been slightly modified to avoid representing a person, from whom we could not obtain the consent form.

Reviewer 3 Report

This paper can only be evaluated with the knowledge that it was written for a special issue. This article is not a research article, so there are no new scientific outcome in the traditional sense. In my opinion, the purpose and result of this paper is a historical review itself. 

The structure of the article is easy to follow, well-written text, interesting reading.  

Author Response

Thanks

Round 2

Reviewer 2 Report

In this revision, the authors have made an effort to address some of the concerns given to them in the review.

They include some paragraphs to contextualize each research topic. This helps to understand the context in which the works described were carried out.

They add some more references.

They extended the “Conclusions and perspectives” section. Now, this section is clearer and more in line with the article's objectives.

The changes made were not very significant, however the article shows improvements over the version previously presented.